# PKN1 promotes synapse maturation by inhibiting mGluR-dependent silencing through neuronal glutamate transporter activation

Hiroki Yasuda [1,2✉], Hikaru Yamamoto[3], Kenji Hanamura[4], Mona Mehruba[5], Toshio Kawamata[6,9], Hiromi Morisaki[7], Masaaki Miyamoto[7], Shinji Takada [8], Tomoaki Shirao[4], Yoshitaka Ono[3,7,10] & Hideyuki Mukai [3,5✉]

Abnormal metabotropic glutamate receptor (mGluR) activity could cause brain disorders; however, its regulation has not yet been fully understood. Here, we report that protein kinase N1 (PKN1), a protein kinase expressed predominantly in neurons in the brain, normalizes group 1 mGluR function by upregulating a neuronal glutamate transporter, excitatory amino acid transporter 3 (EAAT3), and supports silent synapse activation. Knocking out PKN1a, the dominant PKN1 subtype in the brain, unmasked abnormal input-nonspecific mGluR-dependent long-term depression (mGluR-LTD) and AMPA receptor (AMPAR) silencing in the developing hippocampus. mGluR-LTD was mimicked by inhibiting glutamate transporters in wild-type mice. Knocking out PKN1a decreased hippocampal EAAT3 expression and PKN1 inhibition reduced glutamate uptake through EAAT3. Also, synaptic transmission was immature; there were more silent synapses and fewer spines with shorter postsynaptic densities in PKN1a knockout mice than in wild-type mice. Thus, PKN1 plays a critical role in regulation of synaptic maturation by upregulating EAAT3 expression.

[1] Education and Research Support Center, Gunma University Graduate School of Medicine, Maebashi, Gunma 371-8511, Japan. [2] Division of Physiology, Faculty of Medicine, Saga University, Saga, Saga 849-8501, Japan. [3] Biosignal Research Center, Kobe University, Kobe, Hyogo 657-8501, Japan. [4] Department of Neurobiology and Behavior, Gunma University Graduate School of Medicine, Maebashi, Gunma 371-8511, Japan. [5] Graduate School of Medicine, Kobe University, Kobe, Hyogo 650-0017, Japan. [6] Department of Rehabilitation Science, Kobe University Graduate School of Health Sciences, Kobe, Hyogo 654-0142, Japan. [7] Graduate School of Science and Technology, Kobe University, Kobe, Hyogo 657-8501, Japan. [8] Exploratory Research Center on Life and Living Systems, National Institutes of Natural Sciences, Okazaki, Aichi 444-8787, Japan. [9]Deceased: Toshio Kawamata. [10]Deceased: Yoshitaka Ono. ✉email: yasuda@cc.saga-u.ac.jp; mukinase@kobe-u.ac.jp

Neurons in the central nervous system are generated in the embryo, and synaptogenesis starts when pre- and post-synaptic structures meet together. Electrophysiological and anatomical studies proved that there are many silent synapses, which have NMDA receptors (NMDARs) without functional AMPA receptors (AMPARs), in the neonatal hippocampus and the barrel cortex, and the number of synapses with AMPARs increases during development[1–6]. The mechanisms of maturation of synapses remain unclarified. Since both "waking-up" of silent synapses and NMDAR-dependent long-term potentiation (LTP) involve the incorporation of AMPARs at postsynaptic sites, an LTP-like mechanism has been suggested to underlie the maturation of newborn synapses[1,3,6,7]. On the other hand, inhibition of NMDAR function using pharmacological tools or genetic manipulation in the immature brain causes a decrease[8] or an increase in expression of AMPARs[5,9] at synapses, suggesting that developmental expression of synaptic AMPARs is also regulated by an NMDAR-dependent long-term depression (NMDAR-LTD)-like mechanism, which is an activity-dependent decrease in synaptic efficacy and predominant in the immature hippocampus[10,11].

Group 1 metabotropic glutamate receptors (mGluRs) also induce LTD[7,12,13] and have been implicated in brain disorders, including autism, anxiety disorders, and depression[14]. For example, fragile X syndrome is an autism spectrum disorder, caused by abnormality of neural development, and its mouse model shows constitutive activity of mGluR5, a group 1 mGluR[12,14,15]. LTD enhanced by an enhanced mGluR5 function may underlie mental retardation in fragile X syndrome[12,14,15]. On the other hand, mGluR5 activators also improved autistic behaviors in Shank2 knockout (KO) mice, another model for autism[16], suggesting that precise control of mGluR function is essential for normal brain development, although regulation mechanisms for mGluR function are still poorly understood.

Protein kinase N (PKN), which we identified in 1994, is a serine/threonine protein kinase and three subtypes compose the PKN family, which is closely related to the protein kinase C family; however, PKN has a unique regulatory region and is activated by unsaturated fatty acids or Rac and Rho GTPases regulating cytoskeletal organization[17–20]. PKN1 is widely distributed in the entire body of mammals including the brain[21,22]. PKN1 is predominantly expressed in neurons[22] and accounts for 0.01% of total protein in the normal brain[17]. PKN has been implicated in cell proliferation or metastasis of many types of tumor including prostate and bladder cancers, and PKN inhibitors may treat them[20,23]. Very recently, PKN1 has been reported to be involved in axonal outgrowth and presynaptic differentiation of parallel fibers of cerebellar granule cells[24]; however, the physiological roles of PKN in the brain have not been fully clarified yet. Therefore, we investigated the function of PKN1 in the hippocampus by generating PKN1a knockout (KO) mice (Supplementary Fig. 1). PKN1a is a major isoform of PKN1 in the brain (Supplementary Fig. 2). We found that PKN1 normalizes group 1 mGluR function through upregulation of neuronal glutamate transporters. PKN1 masks abnormal mGluR-dependent LTD (mGluR-LTD), prevents silencing of AMPAR synapses, and supports maturation of synapses. We propose that PKN1 is critical for normalizing mGluR activity and essential for normal brain development.

## Results

### PKN1a deletion induces group 1 mGluR-dependent homo- and heterosynaptic LTD in developing hippocampus. We identified two variant PKN1s by molecular cloning. These two variant PKN1s differed in their N-terminal amino acid sequences, which

were hypothesized to arise from alternative splicing (Supplementary Fig. 1). The classical PKN1 (ref. [17]), designated as PKN1a, is expressed predominantly in the brain. We constructed a targeting vector for disrupting PKN1a. PKN1a KO mice carrying the homozygous deletion (PKN1−/−) were viable, born at a frequency expected for Mendelian inheritance, and showed no apparent abnormalities. Biochemical measurement showed that the PKN1 content in the brain of PKN1a KO mice was reduced to ~1/10 in that of wild-type mice (Supplementary Figs. 2 and 3). We confirmed that PKN1 was densely expressed in cell layers both in the CA1 region and the dentate gyrus, and apical dendritelike processes also possessed strong PKN1 signals in the stratum radiatum of the CA1 region in wild-type mice (Supplementary Fig. 2f, g).

Initially, we examined basic synaptic transmission in the CA1 region of developing hippocampal slices in PKN1a KO mice. Synapses in PKN1a KO mice showed a weaker input–output relationship than those in wild-type mice (Fig. 1a; wild type, slopes of regression lines for relationship between fiber volley amplitude and fEPSP slopes, $-2.35 \pm 0.14\,\mathrm{ms^{-1}}$, $n = 10$ from 2 mice; KO, $-1.44 \pm 0.19\,\mathrm{ms^{-1}}$, $n = 10$ from 3 mice; $p = 0.0011$; Welch's $t$-test). The AMPAR/NMDAR ratio was smaller in PKN1a KO mice than in wild-type mice (Fig. 1b; wild type, $0.933 \pm 0.091$, $n = 11$ from 3 mice; KO, $0.642 \pm 0.064$, $n = 15$ from 4 mice; $p = 0.015$; Welch's $t$-test). The frequency of miniature excitatory postsynaptic currents (mEPSCs) was lower in PKN1a KO mice than in wild-type mice (Fig. 1c; wild type, $0.773 \pm 0.066$, $n = 23$ from 4 mice; KO, $0.415 \pm 0.039$, $n = 20$ from 3 mice; $p = 0.000047$; Welch's $t$-test), although the amplitude of mEPSCs was similar between the two groups of mice (Fig. 1c; wild type, $10.24 \pm 0.29$ pA; KO, $9.92 \pm 0.42$ pA). PKN1a did not affect transmitter release because paired-pulse ratios (PPRs) and post-tetanic potentiation were not significantly different between KO and wild-type mice (Fig. 1d). All these electrophysiological properties of synapses lacking PKN1a have been observed in the immature hippocampus[25], suggesting that PKN1 promotes development of synapses. Dendrites possessed fewer (Fig. 2a, b; wild type, $0.898 \pm 0.022$ spines/µm, $n = 30$ from 3 mice; KO, $0.688 \pm 0.026$ spines/µm, $n = 32$ from 3 mice: $p = 0.000000053$; Welch's $t$-test) and approximately 10% smaller spines (Supplementary Fig. 4) in PKN1a KO mice than in wild-type mice. We also classified spines into filopodia, stubby, thin, and mushroom spines, on the basis of spine length and spine head width (see "Methods")[26,27]. Mushroom spines were significantly fewer in KO mice than in wild-type mice (Fig. 2c; wild type, $0.544 \pm 0.021$ spines/µm; KO, $0.382 \pm 0.023$ spines/µm; $p = 0.0000022$; Welch's $t$-test). The decrease in the number of mushroom spines might cause the reduction in averaged spine size in PKN1a KO mice. Interestingly, filopodia were also fewer in KO mice than in wild-type mice (Fig. 2c; wild type, $0.081 \pm 0.011$ spines/µm; KO, $0.043 \pm 0.007$ spines/µm; $p = 0.0041$; Welch's $t$-test). Postsynaptic densities (PSDs) in KO mice were significantly shorter than those in wild-type mice (Fig. 2d; wild type, $0.358 \pm 0.008$ µm, $n = 408$ from 3 mice; KO, $0.225 \pm 0.005$ µm, $n = 440$ from 3 mice; $p = 5.3 \times 10^{-37}$; Welch's $t$-test), indicating that synapses in the hippocampus in PKN1a KO mice are also morphologically abnormal.

We then examined long-term plasticity in the hippocampus. LTP of field excitatory postsynaptic potentials (fEPSPs) was induced normally by high-frequency stimulation and theta-burst stimulation in PKN1a KO mice (Supplementary Fig. 5a, b). Late-phase LTP was not different between wild-type and KO mice (Supplementary Fig. 5c). Thus, the possibility that a deficit in LTP-like mechanisms causes a reduced synaptic transmission in PKN1a KO mice is less likely. However, homosynaptic LTD induced by low-frequency stimulation (LFS) was enhanced in activated synapses in PKN1a KO mice (Fig. 3a, c; wild type,

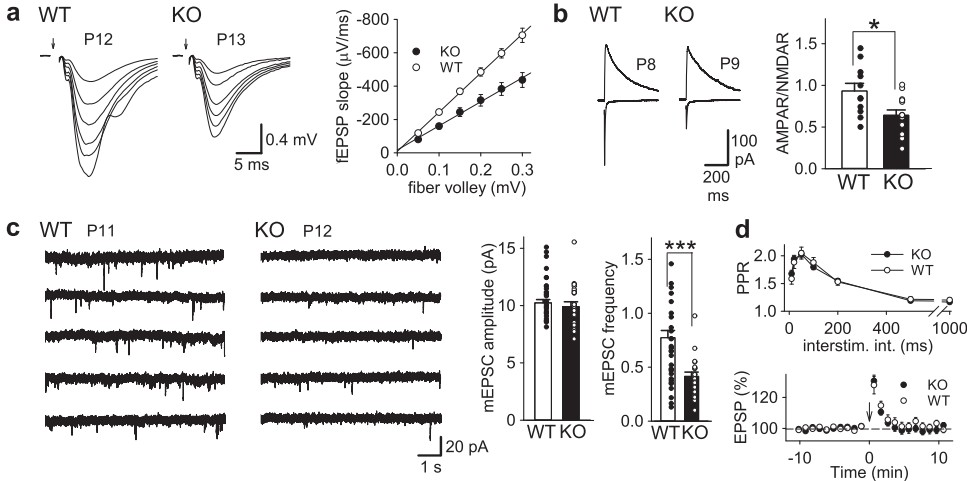

**Fig. 1 AMPAR-mediated synaptic transmission is weakened in PKN1a KO mice. a** Example and summary of input–output relationship in wild type (slopes of regression lines for relationship between fiber volley amplitude and fEPSP slopes, $-2.35 \pm 0.14$ ms$^{-1}$, $n = 10$ from 2 mice at P12–13) and PKN1a KO mice ($-1.44 \pm 0.19$ ms$^{-1}$, $n = 10$ from 3 mice at P12–13; $t_{(17)} = -3.94$, $p = 0.0011$; Welch's $t$-test). **b** Example of AMPAR- and NMDAR-EPSCs and summary of AMPAR/NMDAR ratios (wild type, $n = 11$ from 3 mice at P7–14; KO, $n = 15$ from 4 mice at P7–14; $t_{(17)} = 2.72$, $p = 0.015$; Welch's $t$-test). **c** Example of mEPSCs and summary of mEPSC amplitude (wild type, $n = 23$ from 4 mice at P11–15; KO, $n = 20$ from 3 mice at P11–14) and frequency ($t_{(35)} = 4.64$, $p = 0.000047$; Welch's $t$-test). **d** Summary of paired-pulse ratios (wild type, $n = 15$ from 3 mice; KO, $n = 15$ from 3 mice) and post-tetanic potentiation (wild type, $n = 18$ from 5 mice at P12–14; KO, $n = 17$ from 4 mice at P12–14). All data are shown as mean $\pm$ s.e.m.

$87.7 \pm 2.8\%$ of baseline 50 min after LFS, $n = 17$ from 7 mice; KO, $75.2 \pm 2.3\%$, $n = 22$ from 8 mice; $p = 0.0021$; Welch's $t$-test). Furthermore, heterosynaptic LTD was abnormally induced in inactive synapses in postnatal days 12–18 (P12–18) KO mice (Fig. 3a, c; wild type, $101.8 \pm 2.6\%$; KO, $81.0 \pm 2.9\%$; $p = 0.0000059$; Welch's $t$-test). Homosynaptic LTD in the normal hippocampus induced by single-pulse LFS is mediated by calcium influx through NMDARs, and is not affected by mGluR antagonists[7]. However, homosynaptic LTD in KO mice was inhibited by not only D-APV (50 μM), an NMDAR antagonist, but also LY341495 (100 μM), an mGluR antagonist (Supplementary Fig. 6a and Fig. 3c). The magnitude of NMDAR-dependent homosynaptic LTD that was decreased in KO mice by D-APV (approximately $-20\%$) is comparable to that in the wild type with no drug (Fig. 3c; approximately $-15\%$); therefore, homosynaptic LTD in KO mice consists of normal NMDAR-LTD plus abnormally induced mGluR-LTD. Heterosynaptic LTD in KO mice was not inhibited by D-APV, but suppressed by LY341495 (Supplementary Fig. 6a and Fig. 3c). Homosynaptic LTD was partially inhibited; however, heterosynaptic LTD was not affected by BAPTA (10 mM), a high-affinity calcium chelator that was loaded to neurons through a recording electrode (Supplementary Fig. 6b), suggesting that mGluR-LTD in the hippocampus in PKN1a KO mice is independent of calcium. Furthermore, homosynaptic LTD in KO mice was not affected by LY367385 (100 μM), an mGluR1 antagonist, but was inhibited by MPEP (10 μM), an mGluR5 antagonist (Fig. 3b, c; no drug, $70.9 \pm 2.0\%$, $n = 16$ from 16 mice; LY367385, $71.4 \pm 3.0\%$, $n = 12$ from 7 mice; MPEP, $90.0 \pm 4.3\%$, $n = 13$ from 7 mice; $p = 0.00029$; one-way ANOVA with Tukey–Kramer test). Also, heterosynaptic LTD in KO mice was not affected by LY367385, but was suppressed by MPEP (Fig. 3b, c; no drug, $73.5 \pm 3.8\%$, LY367385, $75.1 \pm 5.5\%$; MPEP, $96.6 \pm 4.2\%$; $p = 0.0013$; one-way ANOVA with Tukey–Kramer test). These results indicate that enhanced synaptically induced spreading LTD induced by single-pulse LFS in PKN1a KO mice is mediated by mGluR5. This is consistent with a lower expression of mGluR1 in CA1 pyramidal cells[28]. mGluR-LTD induced by an agonist or paired-pulse LFS in the normal immature hippocampus is independent of protein

synthesis[29]; however, enhanced mGluR-LTD in immature PKN1 KO mice requires protein translation because a translation inhibitor, cycloheximide (60 μM), was found to suppress mGluR-LTD induced by single-pulse LFS (Fig. 3b, c; homosynaptic pathway, $88.5 \pm 3.7\%$, $n = 10$ from 5 mice, $p = 0.0023$; heterosynaptic pathway, $96.9 \pm 3.7\%$, $p = 0.0028$; one-way ANOVA with Tukey–Kramer test). Next, we applied paired-pulse LFS (1 Hz, 50 ms interstimulus interval, 15 min; PP-LFS) to developing hippocampal slices in the presence of D-APV (50 μM), because low-frequency paired-pulse stimulation induces homosynaptic mGluR-LTD[12,29,30] but not heterosynaptic LTD[30] in the normal hippocampus, and the intensive induction protocol could involve mGluRs other than mGluR5 in PKN1a KO mice. PP-LFS induced homosynaptic LTD (Fig. 4a, b; $87.4 \pm 2.5\%$, $n = 16$ from 7 mice); however, it did not induce heterosynaptic LTD in the P11–16 wild-type hippocampus (Fig. 4a, b; $98.4 \pm 1.8\%$), as reported previously[30]. On the other hand, PP-LFS induced more robust homosynaptic LTD (Fig. 4a, b; $69.9 \pm 2.0\%$, $n = 18$ from 7 mice; $p = 0.0000004$; one-way ANOVA with Tukey–Kramer test) and heterosynaptic LTD (Fig. 4a, b; $80.5 \pm 2.8\%$; $p = 0.00063$; one-way ANOVA with Tukey–Kramer test) in P11–16 PKN1a KO mice. Different from LTD induced by single-pulse LFS, homosynaptic LTD induced by PP-LFS was reduced by an mGluR1 antagonist LY367385 (Fig. 4b; $78.9 \pm 4.6\%$, $n = 5$ from 2 mice). We also tested another mGluR1 antagonist, YM298198 (5 μM), and found that YM298198 has similar effects on homosynaptic LTD (Fig. 4b; $78.7 \pm 2.2\%$, $n = 13$ from 5 mice). These mGluR1 antagonists significantly reduced homosynaptic LTD induced by PP-LFS in the presence of D-APV (Fig. 4a, b; $78.8 \pm 2.0\%$, $n = 18$ from 6 mice, $p = 0.011$; one-way ANOVA with Tukey–Kramer test). An mGluR5 antagonist MPEP also reduced homosynaptic LTD (Fig. 4b; $90.8 \pm 2.2\%$, $n = 15$ from 8 mice), although MPEP and D-APV did not completely inhibit homosynaptic LTD. We also used MTEP (10 μM), another mGluR5 antagonist, and we found that MTEP and D-APV did not completely inhibit homosynaptic LTD in KO mice (Fig. 4b; $85.7 \pm 4.5\%$, $n = 5$ from 3 mice). Thus, mGluR5 antagonists and D-APV significantly reduced homosynaptic LTD (Fig. 4a, b; $89.6 \pm 2.0\%$, $n = 20$ from 11 mice, $p < 0.0000001$; one-way

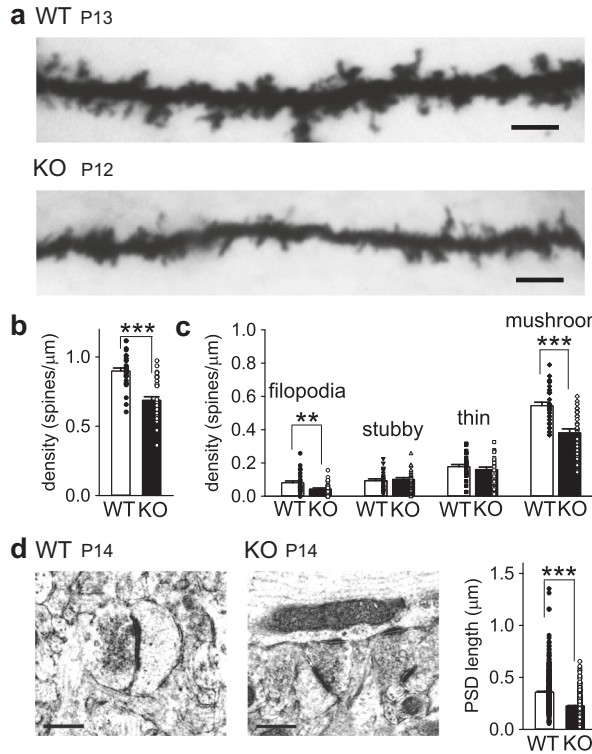

**Fig. 2 PKN1a KO mice have smaller spines with shorter PSDs in the hippocampus than wild-type mice. a** Images of spines on Golgi-stained apical dendrites in CA1 pyramidal neurons in P13 wild type and P12 KO mice. Scale, 5 μm. **b** Summary of spine densities in wild type ($n = 30$ from 3 mice at P12–14) and KO mice ($n = 32$ from 3 mice at P12–14; $t_{(60)} = 6.81$, $p = 0.000000053$; Welch's t-test). **c** Summary of densities of filopodia, stubby, thin, and mushroom spines in wild type and KO mice. Filopodia and mushroom spines were decreased in PKN1a KO mice (filopodia, $t_{(49)} = 3.02$, $p = 0.0041$; mushroom spines, $t_{(60)} = 5.24$, $p = 0.0000022$; Welch's test). **d** Electron microscopy images of PSDs and summary of PSD length in the stratum radiatum of the CA1 region in wild type ($n = 408$ from 3 mice at P12–14) and KO mice ($n = 440$ from 3 mice at P12–14; $t_{(671)} = 13.5$, $p = 5.3 \times 10^{-37}$; Welch's t-test). Scale, 0.4 μm. *$p < 0.05$, **$p > 0.01$, ***$p < 0.001$. All data are shown as mean ± s.e.m.

ANOVA with Tukey–Kramer test). Interestingly, D-APV plus antagonists of mGluR1 (YM298198) and mGluR5 (MPEP or MTEP) did not completely inhibit homosynaptic LTD induced by PP-LFS (Fig. 4a, b; 91.6 ± 1.7%, $n = 18$ from 11 mice (MPEP, $n = 11$ from 7 mice; MTEP, $n = 7$ from 5 mice); significantly different from APV only at $p < 0.0000001$; one-way ANOVA with Tukey–Kramer test), suggesting that PP-LFS activates some mechanisms other than NMDARs and group 1 mGluRs, and induces NMDAR- and group 1 mGluR-independent LTD (see "Discussion"). On the other hand, heterosynaptic LTD induced by PP-LFS in KO mice was inhibited by D-APV plus mGluR1 and mGluR5 antagonists (Fig. 4a, b; 99.0 ± 2.3%, $p = 0.00024$; one-way ANOVA with Tukey–Kramer test), although APV and mGluR1 antagonists, or APV and mGluR5 antagonists did not significantly reduce heterosynaptic LTD (Fig. 4b; APV and mGluR1 antagonists, 88.4 ± 2.7%; APV and mGluR5 antagonists, 89.0 ± 3.1%), suggesting that either of mGluR1 or mGluR5 activity is enough to induce heterosynaptic LTD in PKN1a KO mice. In summary, these results suggest that not only mGluR5 but also mGluR1 is involved in LTD depending on induction protocols in PKN1a KO mice (Table 1). Therefore, PKN1 masks mGluR1- and mGluR5-dependent LTD in the normal developing hippocampus.

**PKN1a suppresses group 1 mGluR-dependent spreading LTD at postsynaptic sites through neuronal glutamate transporter activity.** To examine whether PKN1 downregulates group 1 mGluR function, we used (s)-DHPG, a group 1 mGluR (mGluR1, 5) agonist. LTD induced by DHPG (50 μM) in KO mice was comparable to that in wild-type mice (Supplementary Fig. 7); therefore, simple upregulation of group 1 mGluR function by deleting PKN1 is less likely. Next, we addressed the possibility that an elevated glutamate concentration promotes abnormal mGluR-LTD induction in immature PKN1a KO mice. Previously, reduced glutamate clearance was reported to enhance mGluR-LTD[31]. In the presence of a glutamate transporter antagonist, DL-TBOA at 10 μM, at which concentration glial excitatory amino acid transporter 2 (EAAT2) and neuronal EAAT3 are inhibited[32], homosynaptic LTD was enhanced (Fig. 5a, d; no drug, 79.2 ± 3.3%, $n = 7$ from 6 mice; TBOA, 60.8 ± 3.2%, $n = 10$ from 9 mice; $p = 0.0085$; one-way ANOVA with Tukey–Kramer test) and heterosynaptic LTD was induced in wild-type mice (Fig. 5a, d; no drug, 101.8 ± 4.0%; TBOA, 68.0 ± 3.0%; $p = 0.0000002$; one-way ANOVA with Tukey–Kramer test). Enhanced LTD in the presence of TBOA was blocked by MPEP and LY367385 (Fig. 5a, d; homosynaptic pathway, 88.6 ± 3.2%, $n = 9$ from 9 mice; significantly different from TBOA only at $p = 0.000022$; heterosynaptic pathway, 97.0 ± 3.5%; $p = 0.0000013$; one-way ANOVA with Tukey–Kramer test). Glutamate at extrasynaptic sites, whose concentration could be elevated by TBOA, activates both mGluR1 and mGluR5 (ref. [33]); therefore, LY367385 was also included together with MPEP to block group 1 mGluRs when TBOA was applied. MPEP significantly blocked homosynaptic LTD (Fig. 5b, d; TBOA, 70.3 ± 2.5%, $n = 10$ from 5 mice; TBOA + MPEP, 87.1 ± 4.3%, $n = 12$ from 7 mice; $p = 0.0029$; one-way ANOVA with Tukey–Kramer test), and LY367385 did not affect homosynaptic LTD in the presence of TBOA in wild-type mice (Fig. 5b, d; 71.2 ± 2.3%, $n = 13$ from 8 mice). However, MPEP did not completely inhibit heterosynaptic LTD in the presence of TBOA in wild-type mice (Fig. 5b, d; TBOA, 73.3 ± 1.6%; TBOA + MPEP, 92.5 ± 2.3%; $p = 0.00061$; one-way ANOVA with Tukey–Kramer test), and LY367385 reduced heterosynaptic LTD (Fig. 5b, d; 84.7 ± 4.2%; $p = 0.043$; one-way ANOVA with Tukey–Kramer test). Thus, inhibition of glutamate transporters unmasks mGluR5- and mGluR1-dependent LTD in the developing wild-type hippocampus. TBOA did not enhance LTD in KO mice (Fig. 5c, d; homosynaptic pathway, no drug, 64.4 ± 5.3%, $n = 8$ from 5 mice; TBOA, 66.5 ± 4.4%, $n = 7$ from 5 mice; heterosynaptic pathway, no drug, 72.4 ± 5.3%; TBOA, 69.4 ± 5.0%), suggesting that the effect of TBOA was occluded by PKN1 deletion. Dihydrokainic acid (DHK, 100 μM), a glial EAAT2-selective inhibitor, did not unmask mGluR-LTD in wild-type mice (Fig. 5a, d; homosynaptic pathway, 83.6 ± 3.3%, $n = 14$ from 9 mice; heterosynaptic pathway, 97.4 ± 2.2%). We also found that EAAT3 expression in the developing hippocampus was significantly lower in KO mice than in wild-type mice (Fig. 5e; wild-type mice, 100.9 ± 4.36%, $n = 20$ from 10 mice; KO mice, 80.2 ± 2.7%, $n = 15$ from 8 mice; $p = 0.00035$; Welch's t-test). Furthermore, we examined the effects of PKN1 activity on glutamate uptake through EAAT3 using human SH-SY5Y neuroblastoma cells. Overexpression of wild-type PKN1 did not affect glutamate uptake (Fig. 5f; EAAT3 only, 100.0 ± 1.3%, $n = 21$; EAAT3 + PKN1, 103.1 ± 2.0%, $n = 22$), presumably because endogenous PKN1 in SH-SY5Y cells already activated EAAT3. On the other hand, overexpression of PKN1 without kinase activity (T774A and K644E)[34] reduced EAAT3 glutamate uptake through dominant-negative effects (Fig. 5f; EAAT3 + T774A, 82.7 ± 1.2%, $n = 22$; $p < 0.0000001$; K644E, 85.4 ± 0.9%, $n = 12$; $p = 0.0000007$; one-way ANOVA with Tukey–Kramer test), indicating that endogenous PKN1

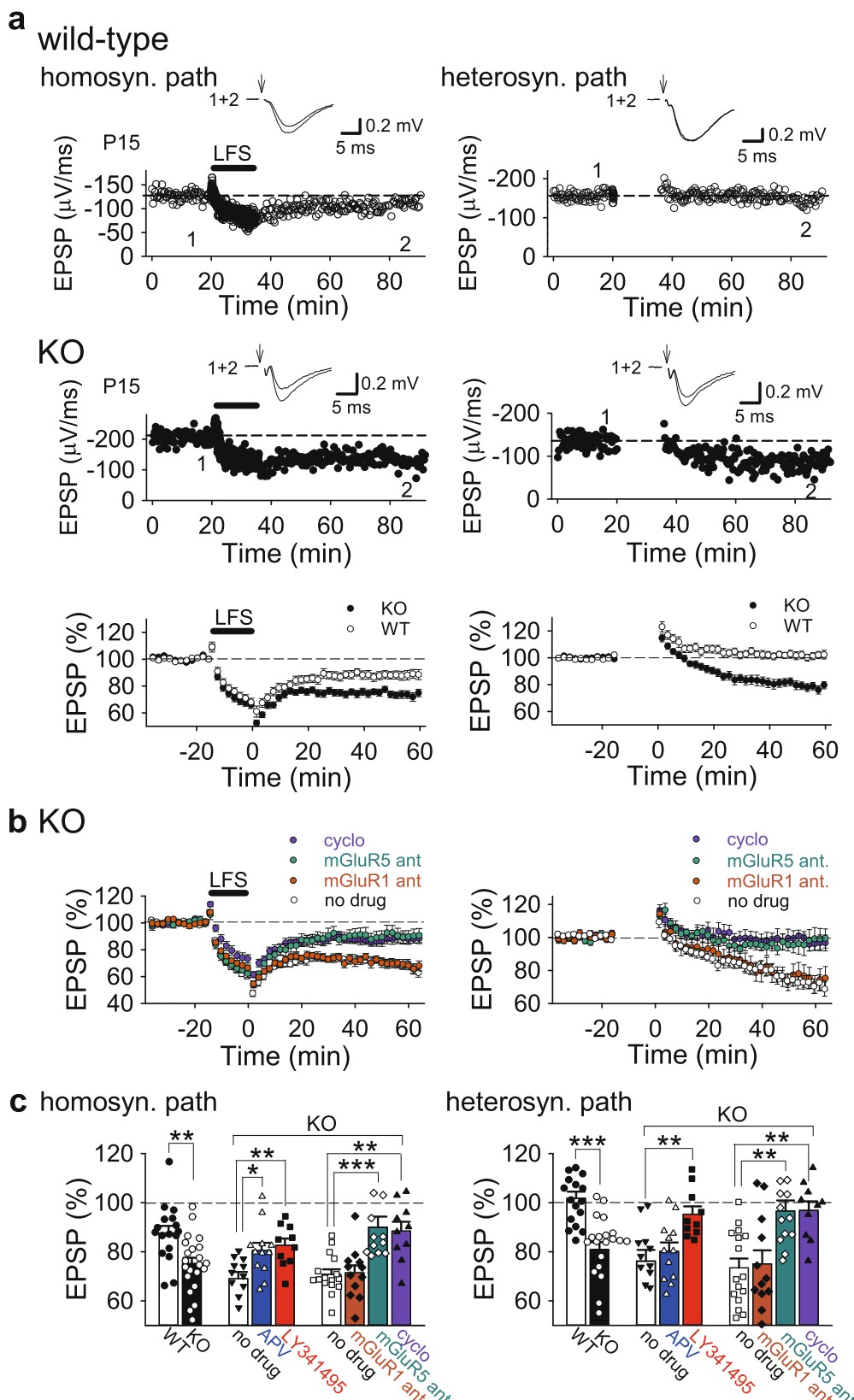

activates EAAT3. Taken together, PKN1 downregulates group 1 mGluR by activating neuronal glutamate transporters and suppresses mGluR-LTD in the normal brain.

We also examined rise time and decay of NMDAR-EPSCs at −70 mV in developing KO mice, because inhibition of neuronal glutamate transporters by soluble amyloid β oligomers was previously shown to prolong the rise time and the decay time constant[31]. We did not find any difference in rise time between wild type and KO mice (Fig. 6a; wild type, 6.16 ± 0.35 ms, $n = 9$ from 3 mice; KO, 6.27 ± 0.35 ms, $n = 11$ from 3 mice). However, NMDAR-EPSC decay time constant was significantly delayed in KO mice compared with wild-type mice (Fig. 6a; wild type,

**Fig. 3 Heterosynaptic LTD is induced by single-pulse LFS in PKN1a KO mice and blocked by an mGluR5 inhibitor. a** Examples and average time course of homo- and heterosynaptic LTD of fEPSPs in P12–18 mice (wild type, $n = 17$ from 7 mice; KO, $n = 22$ from 8 mice). LFS induced more robust homosynaptic LTD ($t_{(34)} = 3.33$, $p = 0.0021$; Welch's $t$-test) and heterosynaptic LTD ($t_{(37)} = 5.28$, $p = 0.0000059$; Welch's $t$-test) in KO mice. **b** Averaged time course of homo- and heterosynaptic LTD in the presence of an mGluR5 antagonist, MPEP, an mGluR1 antagonist, LY367385 (LY), and cycloheximide (cyclo) in PKN1a KO mice at P10–16 (no drug, $n = 16$ from 16 mice; MPEP, $n = 13$ from 7 mice; LY367385, $n = 12$ from 7 mice; cycloheximide, $n = 10$ from 5 mice). MPEP and cycloheximide significantly inhibited homo- ($F_{(3, 47)} = 10.62$; MPEP, $p = 0.00029$; cyclo, $p = 0.0023$; one-way ANOVA with Tukey–Kramer test) and heterosynaptic LTD ($F_{(3, 47)} = 8.89$; MPEP, $p = 0.0013$; cyclo, $p = 0.0028$; one-way ANOVA with Tukey–Kramer test) in KO mice. **c** Summary of the effects of various glutamate receptor antagonists and cycloheximide on homo- and heterosynaptic LTD in PKN1a KO mice. Data of D-APV and LY341495 are from Supplementary Fig. 6a. *$p < 0.05$; **$p < 0.01$; ***$p < 0.001$. All data are shown as mean ± s.e.m.

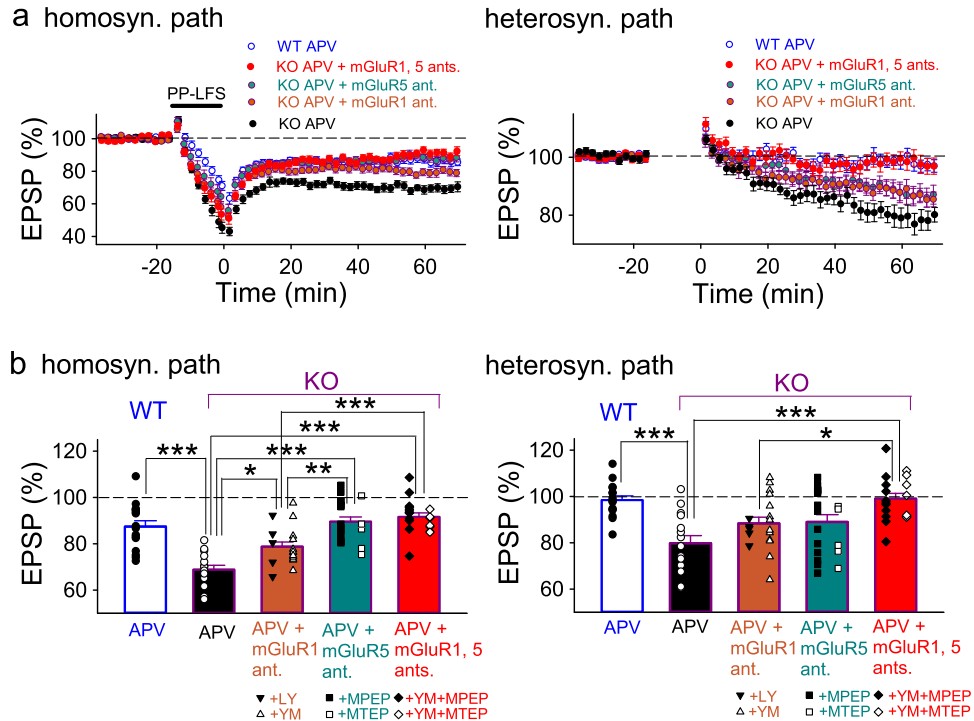

**Fig. 4 Heterosynaptic LTD induced by paired-pulse LFS is inhibited by simultaneous application of mGluR1 and mGluR5 antagonists in PKN1a KO mice. a, b** Average time course (**a**) and summary (**b**) of homo- and heterosynaptic LTD induced by paired-pulse LFS (PP-LFS; 50 ms interstimulus interval, 15 min) in the presence of D-APV in P11–16 PKN1a wild type and KO mice. Homosynaptic LTD induced by PP-LFS is more robust in KO mice ($n = 18$ from seven mice; $F_{(4, 81)} = 19.48$, $p = 0.0000004$; one-way ANOVA with Tukey–Kramer test) than in wild-type mice ($n = 16$ from 7 mice), and heterosynaptic LTD is induced only in KO mice ($F_{(4, 81)} = 7.12$, $p = 0.00063$; one-way ANOVA with Tukey–Kramer test). Different from homosynaptic LTD by single-pulse LFS (Fig. 3b, c), mGluR1 antagonists (LY367385 (LY) and YM298198 (YM)) significantly reduced homosynaptic LTD induced by paired-pulse LFS in KO mice (APV + mGluR1 antagonist $n = 18$ from 6 mice (LY367385, $n = 5$ from 2 mice; YM298198, $n = 13$ from 5 mice); $p = 0.011$; one-way ANOVA with Tukey–Kramer test). mGluR5 antagonists (MPEP and MTEP) also significantly inhibited homosynaptic LTD induced by paried-pulse LFS in KO mice (APV + mGluR5 antagonist, $n = 20$ from 11 mice (MPEP, $n = 15$ from 8 mice; MTEP, $n = 5$ from 3 mice), $p = 0.0000000$; one-way ANOVA with Tukey–Kramer test). Interestingly, D-APV plus mGluR1 and mGluR5 antagonists did not completely inhibit homosynaptic LTD induced by paired-pulse LFS in KO mice (APV + mGluR1 (YM298198) and mGluR5 antagonists, $n = 18$ from 11 mice (MPEP, $n = 11$ from 7 mice; MTEP, $n = 7$ from 5 mice); significantly different from APV only at $p = 0.0000000$; one-way ANOVA with Tukey–Kramer test). Heterosynaptic LTD induced by paired-pulse LFS was inhibited by D-APV plus mGluR1 and mGluR5 antagonists ($p = 0.00024$; one-way ANOVA with Tukey–Kramer test); however, D-APV plus mGluR1 antagonist or D-APV plus mGluR5 antagonist did not significantly reduce heterosynaptic LTD in KO mice. *$p < 0.05$; **$p < 0.01$; ***$p < 0.001$. All data are shown as mean ± s.e.m.

**Table 1 Involvement of mGluR1 and mGluR5 in mGluR-LTD under various conditions.**

|  | Homosynaptic pathway | | Heterosynaptic pathway | |
| --- | --- | --- | --- | --- |
|  | mGluR1 | mGluR5 | mGluR1 | mGluR5 |
| Single-pulse LFS in KO mice | – | ++ | – | ++ |
| Single-pulse LFS in WT mice in TBOA | – | ++ | + | ++ |
| PP-LFS in KO mice | + | ++ | + | + |

Relative involvement of group 1 mGluRs was evaluated using the results shown in Fig. 3b, c for single-pulse LFS in KO mice, Fig. 5b, d for single-pulse LFS in the presence of TBOA in wild-type mice, and Fig. 4 for PP-LFS in KO mice.

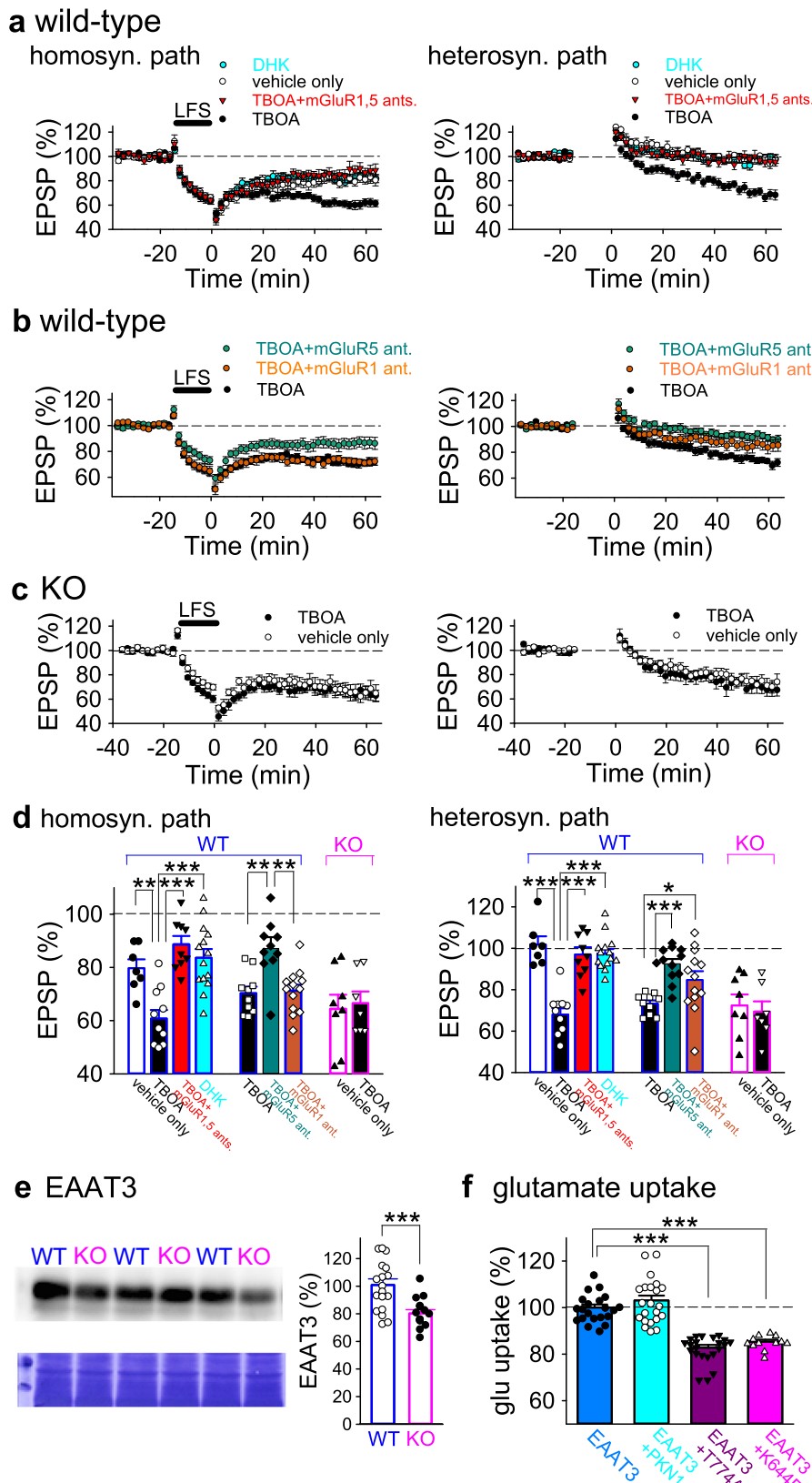

75.9 ± 4.0 ms; KO, 128.6 ± 13.0 ms; $p = 0.0022$; Welch's $t$-test). Developing EAAT3 KO mice have been reported to show a slower NMDAR-ESPC decay; however, prolongation of the rise time has not been mentioned[35]. Therefore, we intracellularly loaded neurons with a high concentration of DL-TBOA (400 µM) through a patch pipette for 10 min, because EAAT3 has a lower

affinity for the intracellular binding site of TBOA than for the extracellular binding site[36]. Intracellular application of TBOA did not affect rise time in both wild type and KO mice (Fig. 6b; wild-type vehicle only (veh.) 6.34 ± 0.26 ms, $n = 8$ from 6 mice; wild-type TBOA, 7.01 ± 0.26 ms, $n = 12$ from 6 mice; KO veh., 6.56 ± 0.35 ms, $n = 9$ from 8 mice; KO TBOA, 6.45 ± 0.52, $n = 11$ from 7

**Fig. 5 Heterosynaptic LTD is induced in wild-type hippocampus in the presence of glutamate transporter inhibitor and blocked by mGluR1 and mGluR5 inhibitors. a** Average time course of homo- and heterosynaptic LTD in the absence of drug ($n = 7$ from 6 mice) and in the presence of 10 µM DL-TBOA (TBOA; $n = 10$ from 9 mice), TBOA plus mGluR5 (MPEP) and mGluR1 (LY367385 (LY)) antagonists ($n = 9$ from 9 mice), and 100 µM dihydrokainic acid (DHK; $n = 14$ from 9 mice) in wild-type mice at P10–16. **b** Average time course of homo- and heterosynaptic LTD in the presence of TBOA only ($n = 10$ from 5 mice), TBOA plus an mGluR5 antagonist MPEP ($n = 12$ from 7 mice), and TBOA plus an mGluR1 antagonist LY ($n = 13$ from 8 mice) in wild-type mice at P11–16. **c** Average time course of homo- and heterosynaptic LTD in the absence of drug ($n = 8$ from 5 mice) and in the presence of TBOA ($n = 7$ from 5 mice) in PKN1a KO mice at P11–14. **d** Summary of LTD experiments. The percentage of fEPSPs 55–60 min after LFS is shown. LFS induced enhanced homo- ($F_{(3, 36)} = 11.89$, $p = 0.0085$; one-way ANOVA with Tukey–Kramer test) and heterosynaptic LTD ($F_{(3, 36)} = 23.82$, $p = 0.0000002$; one-way ANOVA with Tukey–Kramer test) in wild-type mice in the presence of 10 µM DL-TBOA, which blocks glial EAAT2 and neuronal EAAT3 at this concentration. The enhancement was inhibited by simultaneous application of mGluR1 and mGluR5 antagonists (homosynaptic path, $p = 0.000022$; heterosynaptic path, $p = 0.0000013$; one-way ANOVA with Tukey–Kramer test). Enhanced homosynaptic LTD was significantly inhibited by MPEP ($F_{(2, 32)} = 8.83$, $p = 0.0029$; one-way ANOVA with Tukey–Kramer test), but not by LY367385. However, heterosynaptic LTD is reduced but not completely blocked by MPEP ($F_{(2, 32)} = 8.73$, $p = 0.00061$; one-way ANOVA with Tukey–Kramer test), and partially inhibited by LY367385 ($p = 0.043$; one-way ANOVA with Tukey–Kramer test). In all, 100 µM DHK, an EAAT2-selective inhibitor, did not unmask mGluR-LTD in wild-type mice. **e** Examples (wild type, P12; KO, P11) and summary of quantitative immunoblot analyses of hippocampal EAAT3 expression in wild type ($n = 20$ from 10 mice at P12–15) and KO mice ($n = 15$ from 8 mice at P11–16). EAAT3 expression was significantly lower in KO mice than in wild-type mice ($t_{(31)} = 4.02$, $p = 0.00035$; Welch's t-test). Original images are presented in Supplementary Fig. 8. **f** Summary of the effects of overexpression of wild-type PKN1 and PKN1 without kinase activity (T774A, K644E) on glutamate uptake through exogenous EAAT3 in human SH-SY5Y neuroblastoma cells (control, $n = 21$; PKN1, $n = 22$; T774A, $n = 22$, $F_{(3, 73)} = 44.03$, $p < 0.0000001$; K644E, $n = 12$, $p = 0.0000007$; one-way ANOVA with Tukey–Kramer test). *$p < 0.05$; **$p < 0.01$, ***$p < 0.001$. All data are shown as mean ± s.e.m.

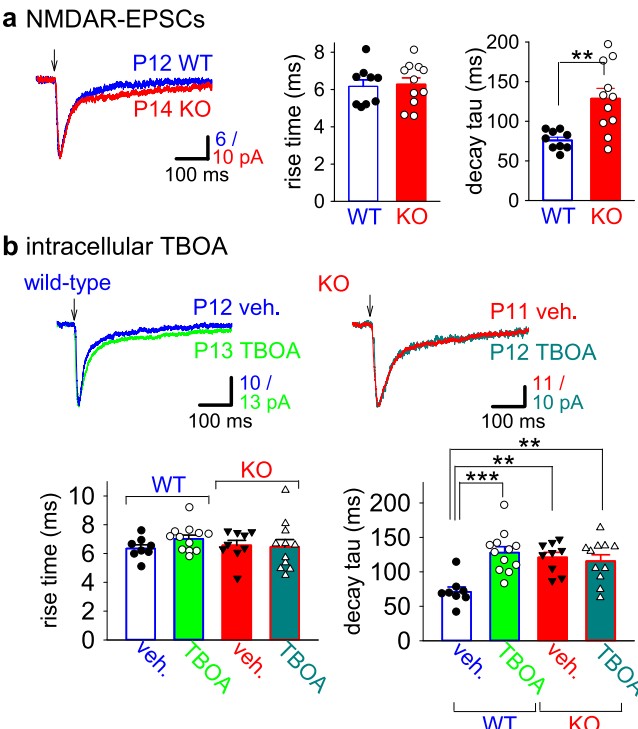

**Fig. 6 Decay of NMDAR-ESPCs is prolonged in PKN1a KO and TBOA-loaded neurons. a** Summary of 10–90% rise time and decay time constant of NMDAR-EPSCs at −70 mV in a low-Mg$^{2+}$ ACSF in P12–14 wild type ($n = 9$ from 3 mice) and PKN1a KO mice ($n = 11$ from 3 mice). Rise time was not different between wild type and KO mice; however, decay time constant in KO mice was significantly longer than that in wild-type mice ($t_{(12)} = −3.88$, $p = 0.0022$; Welch's t-test). **b** Summary of the effects of intracellularly loaded TBOA on 10–90% rise time and decay time constant of NMDAR-ESPCs in P11–15 wild type (vehicle, $n = 8$ from 6 mice; TBOA, $n = 12$ from 6 mice) and KO (vehicle, $n = 9$ from 8 mice; TBOA, $n = 11$ from 7 mice) mice. Intracellular loading of TBOA did not affect rise time in both wild type and KO mice. TBOA prolonged decay time constant in wild-type mice ($F_{(3, 36)} = 7.97$, $p = 0.00026$); however, TBOA did not affect NMDAR-ESPC decay in PKN1a KO mice. **$p < 0.01$, ***$p < 0.001$. All data are shown as mean ± s.e.m.

mice). However, intracellular TBOA significantly prolonged decay time constant of NMDAR-EPSCs in wild-type mice (Fig. 6b; veh., $70.9 ± 7.2$ ms; TBOA, $127.9 ± 8.6$ ms; $p = 0.00026$; one-way ANOVA with Tukey–Kramer test). On the other hand, TBOA did not change the decay time constant in PKN1a KO mice (Fig. 6b; veh., $120.9 ± 7.4$ ms; TBOA, $115.5 ± 9.3$ ms), presumably because EAAT3 activity is reduced in KO mice and the effect of TBOA on EAAT3 in KO mice is occluded. Thus, knocking out PKN1a and intracellular application of TBOA have the same effects on NMDAR-EPSC kinetics in the developing hippocampus, suggesting that neuronal glutamate transporter activity is reduced in the hippocampus of PKN1a KO mice.

Next, we characterized the expression of spreading LTD in the immature hippocampus in PKN1a KO mice using whole-cell recordings. We confirmed that homosynaptic LTD was enhanced and heterosynaptic LTD was induced in PKN1a KO mice in whole-cell current-clamp recordings (Fig. 7a, b; homosynaptic pathway, wild type, $90.5 ± 6.3\%$ 60 min after LFS; $n = 16$ from 14 mice; KO, $72.1 ± 6.4\%$ $n = 20$ from 10 mice, $p = 0.048$; heteosynaptic pathway, wild type, $101.1 ± 4.1\%$; KO, $66.9 ± 6.8\%$; $p = 0.00015$; Welch's t-test). PPRs did not significantly increase 60 min after LFS (Fig. 7b; homosynaptic pathway, $1.12 ± 0.07$, $n = 9$ from 6 mice; heterosynaptic pathway, $1.09 ± 0.13$), indicating that no significant attenuation of presynaptic transmitter release is involved in LTD in KO mice. Then, we examined where PKN1 regulates mGluR-LTD. Initially, we loaded a selective PKN inhibitor peptide, PRL or its control peptide, DNER[37], into a CA1 pyramidal neuron in wild-type mice through a patch pipette at a concentration of 500 µM. PRL loaded into postsynaptic neurons unmasked enhanced and spreading LTD in wild-type mice but its control peptide DNER did not (Fig. 7c; homosynaptic pathway, PRL $72.5 ± 4.5\%$, $n = 17$ from 11 mice; DNER, $89.1 ± 4.8\%$, $n = 15$ from 11 mice; $p = 0.017$; heterosynaptic pathway, PRL $76.2 ± 9.0\%$; DNER, $107.1 ± 6.0\%$; $p = 0.0083$; Welch's t-test), suggesting that PKN1 inhibits LTD at postsynaptic sites. Heterosynaptic LTD in PKN1a KO mice was occluded by the two preceding LFSs (Supplementary Fig. 9a), suggesting that heterosynaptic LTD shares some common expression mechanisms with homosynaptic LTD in PKN1a KO mice. Interestingly, homosynaptic LTD was not completely occluded by heterosynaptic LTD (Supplementary Fig. 9b). We consider that small LTD induced by third LFS in heterosynaptic

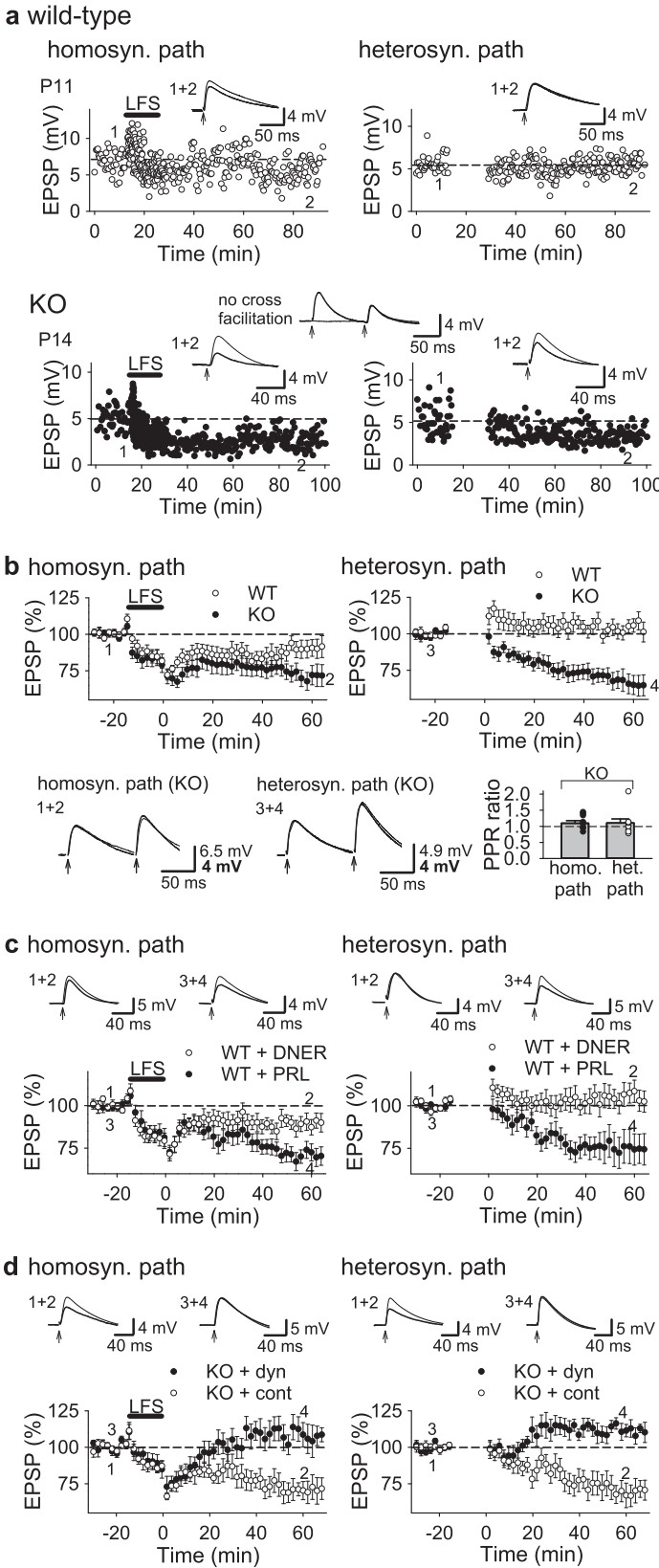

pathway is NMDAR-LTD, because NMDAR-LTD is not occluded by mGluR-LTD[38]. We also tested the effects of an inhibitor of endocytosis on homo- and heterosynaptic LTDs in PKN1a KO mice. Dynamin inhibitory peptide (2 mM) loaded into neurons completely blocked both homo- and heterosynaptic LTD in

PKN1a KO mice; however, its scramble peptide did not (Fig. 7d; homosynaptic pathway, scrambled peptide, 70.9 ± 5.3%, $n = 12$ from 7 mice; inhibitory peptide, 110.7 ± 8.7%, $n = 9$ from 7 mice; $p = 0.0015$; heterosynaptic pathway, scrambled peptide, 67.9 ± 6.9%; inhibitory peptide, 111.0 ± 4.9%; $p = 0.000066$; Welch's

**Fig. 7 Inhibition of PKN1 and endocytosis in postsynaptic neurons unmasks homo- and heterosynaptic LTD. a** Sample experiments illustrating homo- (left) and heterosynaptic LTD of EPSPs (right) in P11 wild type (upper) and P14 PKN1a KO mice (lower) in whole-cell recordings. **b** Summary of homo- (left; wild type, $n = 16$ from 14 mice; KO, $n = 20$ from 10 mice; $t_{(34)} = 2.05$, $p = 0.048$; Welch's $t$-test) and heterosynaptic LTD of EPSPs (right; $t_{(30)} = 4.33$, $p = 0.00015$; Welch's $t$-test) in P10–16 mice. No significant difference was observed between PPRs at baseline and 60 min after LFS in homo- and heterosynaptic pathways in KO mice (lower graph; $n = 9$ from 6 mice). **c** Summary of the effects of postsynaptic injection of a PKN inhibitor peptide, PRL, and a control peptide, DNER, on homo- (left; PRL, $n = 17$ from 11 mice; DNER, $n = 15$ from 11 mice; $t_{(30)} = -2.53$, $p = 0.017$; Welch's $t$-test) and heterosynaptic LTD (right; $t_{(27)} = -2.85$, $p = 0.0083$; Welch's $t$-test) in P11–16 wild-type mice. **d** Summary of the effects of postsynaptic injection of dynamin inhibitor peptide (dyn) and its scrambled peptide (cont) on homo- (left; dyn, $n = 9$ from 7 mice, cont, $n = 12$ from 7 mice; $t_{(14)} = -3.91$, $p = 0.0015$; Welch's $t$-test) and heterosynaptic LTD (right; $t_{(19)} = -5.08$, $p = 0.000066$; Welch's $t$-test) in P12–18 PKN1a KO mice. All data are shown as mean ± s.e.m.

$t$-test). Taken together, PKN1 suppresses mGluR5-dependent LTD that is mediated by endocytosis at postsynaptic sites.

**PKN1 supports maturation of synapses by inhibiting silencing of AMPAR synapses.** In the normal immature hippocampus, LTD is not associated with generation of silent synapses; the mean amplitude of successful EPSCs is reduced but failure rate does not increase after LFS in minimal stimulation experiments[39,40]. We examined whether silent synapses are generated by LFS in developing PKN1a KO mice. After stable control responses with minimal stimulation for at least 10 min, low-frequency pairing stimulation (1 Hz 5 min paired with −50 mV 150 ms pulses) induced modest homosynaptic LTD (Fig. 8a, c; 77.3 ± 8.3% of baseline 30 min after pairing, $n = 9$ from 8 mice) and no heterosynaptic LTD (Fig. 8a, c; 128.5 ± 24.6%, $n = 8$ from 7 mice) in wild-type mice. No significant changes in failure rate were detected in homosynaptic (Fig. 8d; baseline, 0.212 ± 0.036; 30 min after pairing, 0.272 ± 0.034; +40 mV, 0.189 ± 0.036) or heterosynaptic (Fig. 8d; baseline, 0.276 ± 0.035; 30 min after pairing, 0.225 ± 0.056; +40 mV, 0.225 ± 0.047) pathway. However, the exact same stimulation induced more robust homosynaptic LTD (Fig. 8b, c; 51.7 ± 8.3%, $n = 15$ from 14 mice; significantly smaller than that in the wild type at $p = 0.041$; Welch's $t$-test) and heterosynaptic LTD (Fig. 8b, c; $n = 15$ from 14 mice, 64.8 ± 11.9%; $p = 0.042$; Welch's $t$-test) in KO mice. The failure rate increased 30 min after pairing in homosynaptic (Fig. 8d; baseline, 0.269 ± 0.040; 30 min after pairing, 0.466 ± 0.044, $p = 0.0001$; one-way repeated-measures ANOVA with Holm's post hoc test) and heterosynaptic (Fig. 8d; control, 0.311 ± 0.038; 30 min, 0.473 ± 0.058; $p = 0.00033$; one-way repeated-measures ANOVA with Holm's post hoc test) pathways in KO mice. The failure rate at +40 mV was smaller than those at baseline and 30 min after pairing at −70 mV in homosynaptic (Fig. 8d; +40 mV, 0.142 ± 0.035; significantly smaller than baseline at $p = 0.019$ and 30 min at $p = 0.00000058$; one-way repeated-measures ANOVA with Holm's post hoc test) and heterosynaptic (Fig. 8d; +40 mV, 0.174 ± 0.024; significantly smaller than baseline at $p = 0.0067$ and 30 min at $p = 0.00025$; one-way repeated-measures ANOVA with Holm's post hoc test) pathways in KO mice, suggesting that LTD was associated with an increase in silent synapses only in PKN1a KO mice.

Thus far, we found that synapses lacking PKN1 can be easily depressed and even silenced by weak activity in the immature hippocampus. Then, what is the physiological consequence of PKN knockout in vivo? We investigated the percentage of silent synapses among naive hippocampal synapses. In minimal stimulation experiments, the failure rate at −70 mV was significantly different from that at +40 mV in wild-type mice (Fig. 9; −70 mV, 0.402 ± 0.041; +40 mV, 0.287 ± 0.041; $n = 17$ from 8 mice; $p = 0.0034$; paired $t$-test). However, the failure rate at −70 mV was much higher than that at +40 mV in KO mice (Fig. 9; −70 mV, 0.685 ± 0.056; +40 mV, 0.318 ± 0.069, $n = 12$ from 5 mice; $p = 0.000045$; paired $t$-test), indicating that the hippocampus in PKN1a KO mice has much more silent synapses than that in wild-type mice (Fig. 9b; wild type, 28.1 ±

5.4%; KO, 67.6 ± 6.0%; $p = 0.000050$; Welch's $t$-test). Therefore, PKN1 supports "wake-up" of silent synapses by suppressing AMPAR synapse silencing.

## Discussion

Group 1 mGluRs mediate LTD and are implicated in many brain disorders[12,14,15,41]; however, regulation of their action has remained unclarified. We developed PKN1a KO mice and found that deleting PKN1a unmasks group 1 mGluR-dependent input-nonspecific LTD in the developing CA1 region. PKN1a elevates glutamate uptake activity of EAAT3, a neuronal glutamate transporter. Input-nonspecific mGluR-LTD is mimicked by a glutamate transporter inhibitor, TBOA, in wild-type mice, and the effects of TBOA are occluded in KO mice. These data indicate that PKN1 upregulates neuronal glutamate transporters and normalizes group 1 mGluR function (Fig. 9c). Furthermore, LFS induced silencing of AMPAR synapses only in PKN1a KO mice, and the hippocampus in PKN1a KO mice possesses more silent synapses, which is a characteristic feature of the immature brain[1–5]. Thus, PKN1 supports maturation of synapses through inhibition of LTD-like mechanisms by upregulating neuronal glutamate transporters.

**PKN1 upregulates glutamate transporters and normalizes mGluR activity.** In a mouse model of fragile X syndrome, mGluR5s are constitutively active and account for phenotypes including excessive mGluR-LTD[12,15]. LFS induced abnormal mGluR5- and translation-dependent LTD in PKN1a KO mice (Fig. 3). However, DHPG-induced LTD in KO mice was comparable to that in wild-type mice (Supplementary Fig. 7), indicating that PKN1 does not generally downregulate group 1 mGluR function. Rather, PKN1 upregulates activity of neuronal glutamate transporters, reduces glutamate concentration and prevents abnormal mGluR-LTD (Fig. 9c). EAAT3 is modestly expressed mainly at perisynaptic sites in hippocampal neurons[42] and has less capacity of glutamate clearance than glial EAAT2 (ref. [43]). However, knocking out PKN1a reduced EAAT3 expression and PKN1 inhibition downregulated EAAT3 activity (Fig. 5e, f). mGluR-LTD was also induced in wild-type mice in the presence of 10 μM DL-TBOA. Ten micromolar DL-TBOA inhibits EAAT2 and EAAT3 but not EAAT1 because the $IC_{50}$ values of DL-TBOA for EAAT1, EAAT2, and EAAT3 are 70, 6, and 6 μM, respectively[32]. mGluR-LTD was not induced in the presence of an EAAT2-selective inhibitor, DHK (Fig. 5a, d). These data indicate that reduced EAAT3 activity is responsible for mGluR-LTD in wild-type mice in the presence of TBOA. Many synapses are not covered by glial processes in the hippocampus[44,45], so EAAT3 is likely to uptake glutamate at synapses more effectively than EAAT2 and downregulates synaptic group 1 mGluR activity at least in the developing hippocampus where astrocytes are immature[45]. mGluR1 and mGluR5 are located at perisynaptic sites[28,46]; therefore, presumably a higher glutamate concentration is needed to activate group 1 mGluRs than it is to activate

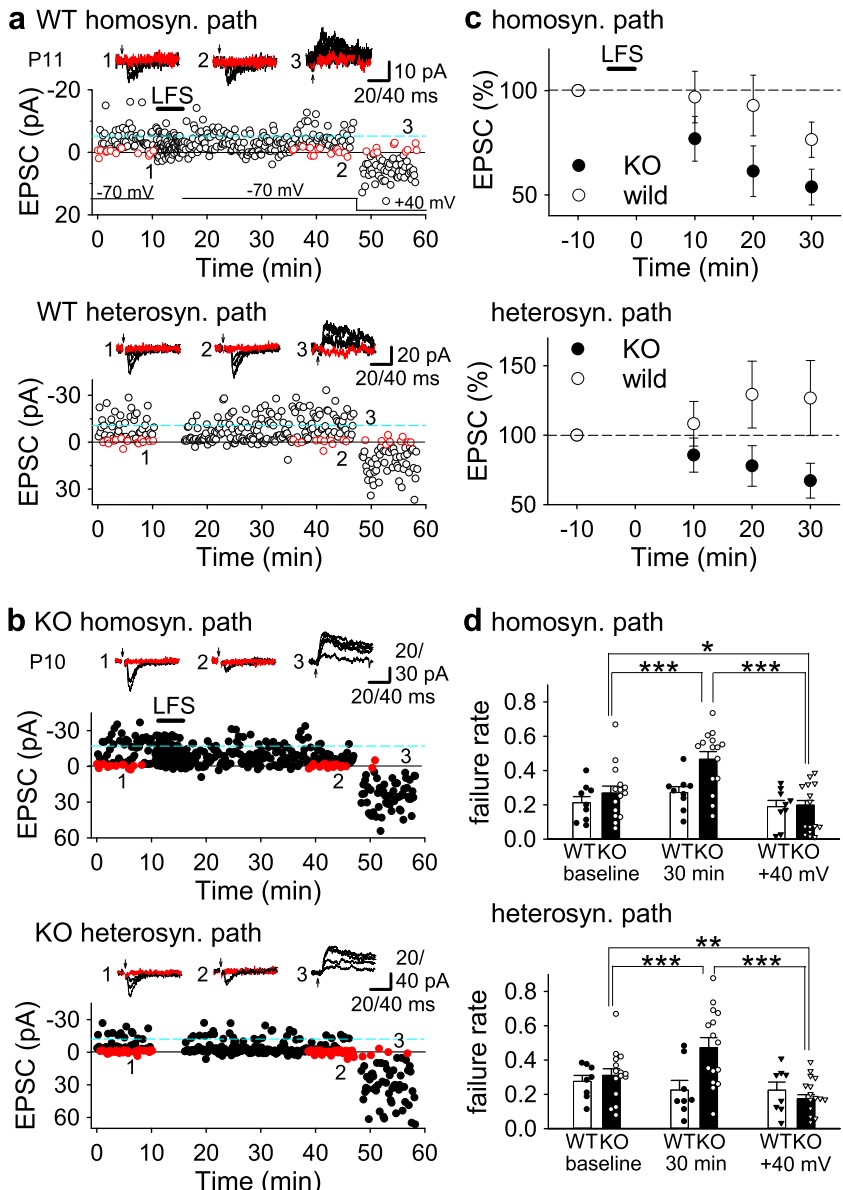

**Fig. 8 LTD is associated with generation of silent synapses in PKN1a KO mice. a, b** Examples of LTD induced by mild low-frequency pairing protocol (LFS, 1 Hz 5 min stimulation paired with −50 mV 150 ms pulses) in minimal stimulation experiments in P11 wild type (**a**) and P10 PKN1a KO mice (**b**). Symbols and traces in red in graphs and EPSC samples indicate failure events. **c** Summary of LTD in homo- (upper; wild type, 77.3 ± 8.3% 30 min after pairing, $n = 9$ from 8 mice; KO, 51.7 ± 8.3%, $n = 15$ from 14 mice; $t_{(20)} = 2.18$, $p = 0.041$; Welch's $t$-test) and heterosynaptic pathways (lower; wild type, 128.5 ± 24.6%, $n = 8$ from 7 mice; KO, 64.8 ± 11.9%, $n = 15$ from 14 mice; $t_{(10)} = 2.33$, $p = 0.042$; Welch's $t$-test). Wild-type mice were at P8–11 and KO mice were at P8–12. **d** Summary of failure rate in homo- (upper) and heterosynaptic (lower) pathways in LTD experiments. The pairing protocol did not induce significant changes in failure rate between baseline, 30 min after pairing at −70 mV and at +40 mV in homo- and heterosynaptic pathways in wild-type mice. However, failure rate 30 min after pairing was significantly higher than that at baseline in homosynaptic ($F_{(2, 28)} = 38.41$, $p = 0.0001$; one-way repeated-measures ANOVA with Holm's post hoc test) and heterosynaptic ($F_{(2, 28)} = 23.76$, $p = 0.00033$; one-way repeated-measures ANOVA with Holm's post hoc test) pathways in PKN1a KO mice. Also, failure rate at +40 mV was significantly lower than that at baseline (homosynaptic pathways, $p = 0.019$; heterosynaptic pathways, $p = 0.0067$; one-way repeated-measures ANOVA with Holm's post hoc test) and that 30 min after pairing at −70 mV (homosynaptic pathways, $p = 0.00000058$; heterosynaptic pathways, $p = 0.00025$; one-way repeated-measures ANOVA with Holm's post hoc test). *$p < 0.05$; **$p < 0.01$; ***$p < 0.001$. All data are shown as mean ± s.e.m.

NMDARs, which are located in PSDs just below presynaptic active zones, and mGluR-LTD is efficiently induced in PKN1a KO mice in which EAAT3 activity is reduced.

Previous anatomical studies suggest that CA1 pyramidal cells mainly have mGluR5 and minimally express mGluR1 (ref. [28,46]). However, more recent electrophysiological studies showed that mGluR1 activation is involved in group 1 mGluR-dependent LTD[33,47] and regulation of LTP induction[48] in the CA1 region of

the hippocampus. We presume that mGluR5 is more abundantly expressed than mGluR1, but mGluR1 is still functional at perisynaptic sites on the CA1 pyramidal neurons. Therefore, single-pulse LFS did not activate mGluR1 (Fig. 3b, c and Table 1) and a more intensive PP-LFS elevated glutamate concentration at synaptic clefts and activated perisynaptic mGluR1 as well as mGluR5. However, mGluR1 was found to be involved in homo- and heterosynaptic LTD induced by PP-LFS in PKN1a KO mice

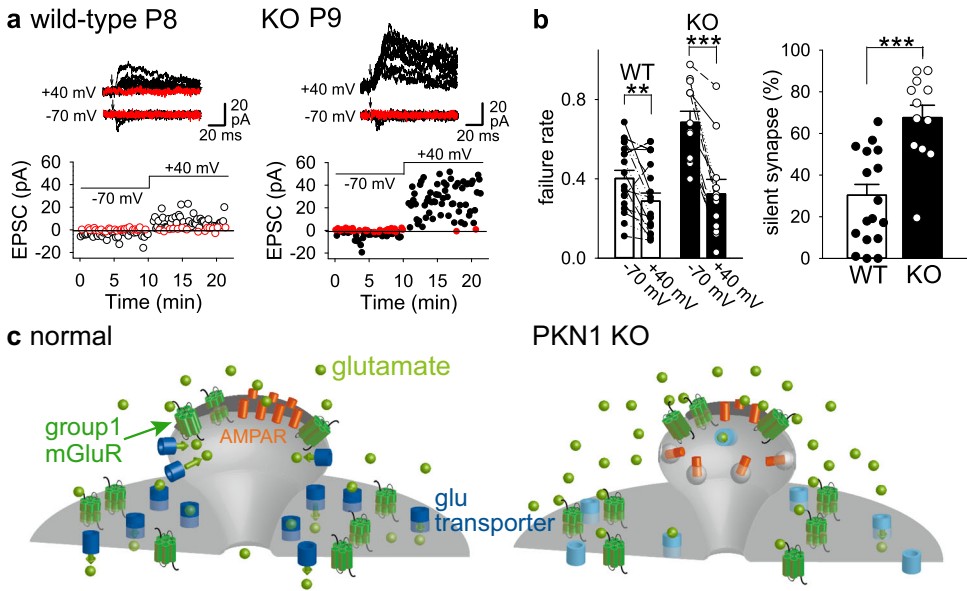

**Fig. 9 PKN1 KO mice possesses more silent synapses than wild-type mice. a** Example of EPSCs at −70 and +40 mV in minimal stimulation experiments in naïve P8 wild type and P9 KO slices. **b** Summary of failure rate (wild type, −70 mV, 0.402 ± 0.041, +40 mV, 0.287 ± 0.041; $n = 17$ from 8 mice; $t_{(16)} = 3.42$, $p = 0.0034$; KO, −70 mV, 0.685 ± 0.056, +40 mV, 0.318 ± 0.069, $n = 12$ from 5 mice; $t_{(11)} = 6.48$, $p = 0.000045$; paired $t$-test) and percentage of silent synapses (wild type, 28.1 ± 5.4%; KO, 67.6 ± 6.0%; $t_{(25)} = -4.88$, $p = 0.000050$; Welch's $t$-test) in naïve P8–12 wild type and P8–12 KO slices. **c** Schematic illustration of spines in the hippocampus of wild type and PKN1a KO mice. PKN1 suppresses group 1 mGluR-dependent LTD by upregulating expression of neuronal glutamate transporters in normal developing synapses and AMPARs remain at postsynaptic sites (left). In PKN1 KO mice, reduction of neuronal glutamate transporter expression causes group 1 mGluR activation. mGluR-dependent LTD-like mechanisms induce endocytosis of AMPARs at postsynaptic sites and increase silent synapses, which is a characteristic feature of the immature hippocampus (right).

(Fig. 4 and Table 1). We also assume that glutamate concentration during single-pulse LFS in the presence of TBOA in wild-type mice might be higher than that during single-pulse LFS in KO mice and lower than that during PP-LFS in KO mice, because mGluR1 is involved in heterosynaptic LTD but not in homosynaptic LTD in the presence of TBOA in wild-type mice (Fig. 5b, d and Table 1). Very interestingly, D-APV plus antagonists of mGluR1 (YM298198) and mGluR5 (MPEP or MTEP) did not completely inhibit homosynaptic LTD induced by PP-LFS in KO mice (Fig. 4a, b). LTD induced by PP-LFS in the normal rat hippocampus is not inhibited by APV + 20 μM LY341495, LY367385, and MPEP, but it is suppressed by APV + 100 μM LY341495 (ref. [33]). LY341495 inhibits group 2 and/or group 3 mGluRs with $K_d$ or $K_i$ values of less than 0.1 μM except for mGluR4, the $K_i$ value for which is 22.0 μM[49]. Twenty micromolar LY341495 may not completely inhibit mGluR4 and it may induce LTD in the presence of APV + 20 μM LY341495, LY367385, and MPEP. Therefore, mGluR4 might also be involved in LTD induced by PP-LFS in PKN1a KO mice.

So far a few reports have argued that neuronal glutamate transporter dysfunction facilitates induction of NMDAR-LTD[31] or internalization of AMPARs[44]. mGluR-dependent LTD was induced only when fewer pulses of LFS were applied in the presence of amyloid-β proteins, which inhibit neuronal glutamate transporters[31]. We are not sure why NMDAR-LTD was not enhanced in PKN1a KO mice in our hands. Weak stimulation prefers induction of mGluR-LTD[38], and we always record smaller fEPSPs for baseline (approximately 0.2–0.4 mV, see Fig. 3). Our experimental conditions might be appropriate for mGluR-LTD rather than NMDAR-LTD.

PKN1 deletion unmasked mGluR5-dependent heterosynaptic LTD (Fig. 3). There are few reports on heterosynaptic LTD[50,51] and induction mechanisms are poorly understood. Intracellular calcium mediates homosynaptic LTD[7], and spread of calcium is a candidate signal for induction of heterosynaptic LTD[51]. In contrast,

heterosynaptic LTD in PKN1a KO mice is independent of calcium (Supplementary Fig. 6b). mGluR-LTD in the normal immature hippocampus does not require protein translation[29]; however, abnormal mGluR-LTD in immature PKN1 KO mice is translation-dependent. Therefore, PKN1 might restrict translation and diffusion of produced proteins to inhibit spread of LTD.

**PKN1 supports maturation of synapses through EAAT3 activity during development.** Single-pulse LFS elicits only modest NMDAR-LTD in the normal hippocampus[52,53]. However, LFS induced NMDAR-LTD and additional mGluR5-dependent LTD that spread also to inactive synapses (Fig. 3) and generated silent synapses in the developing hippocampus lacking PKN1a (Fig. 8). Paired-pulse LFS induced mGluR1- and mGluR5-dependent homo- and heterosynaptic LTD in KO mice (Fig. 4). On the other hand, three different types of induction protocol induced almost the same amount of LTP in both wild-type and PKN1a KO mice (Supplementary Fig. 5). Therefore, we consider that a group 1 mGluR-dependent LTD-like mechanism induces much more silent synapses in in vivo PKN1a KO mice (Fig. 9), which resulted in the attenuated input–output relationship (Fig. 1a), a lower AMPAR/NMDAR ratio (Fig. 1b), a lower mEPSC frequency (Fig. 1c), and a smaller number of spines (Fig. 2). These are all the features of young synapses[1–5,25]. Previously, not only NMDAR-LTD[54] but also mGluR-LTD[55,56] was suggested to induce shrinkage and retraction of spines. Furthermore, shrinkage of large spines requires group 1 mGluR activation[56]. Therefore, mature mushroom spines might be preferentially shrunk and reduced by mGluR-LTD in PKN1a KO mice (Fig. 2c). Interestingly, filopodia were also decreased; however, immature stubby and thin spines were not changed in PKN1a KO mice (Fig. 2c). LTP is intact in PKN1a KO mice (Supplementary Fig. 5), so LTP-like mechanisms might convert filopodia into immature stubby and thin spines, but enhanced mGluR-LTD might reduce further maturation of spines. Many mGluRs are expressed at perisynaptic

sites[28]. Because PKN1a KO mice have smaller spines (Supplementary Fig. 4) with shorter PSDs (Fig. 2d) than wild-type mice, mGluRs could be physically closer to presynaptic active zones. The possible closeness of group 1 mGluRs to active zones might also help induction of mGluR-LTD in PKN1a KO mice.

EAAT3 is modestly expressed in hippocampal neurons and has less capacity for glutamate clearance than EAAT2 (ref. [43]), and its physiological roles have not been well elucidated. In the cerebellum, neuronal glutamate transporters (EAAT3 and EAAT 4) distribute closely to mGluRs at postsynaptic sites and negatively regulate mGluR-LTD[57]. mGluR5 colocalizes with EAAT3 also in the CA1 region of the hippocampus[58]. Furthermore, EAAT3 expression precedes that of glial glutamate transporters in the first postnatal week, peaks in the second postnatal week, and decreases thereafter in the brain of rodents[45,59]. Considering that many synapses are not covered by glial processes in the hippocampus[44,45], EAAT3 may be more important in synapses specifically in the developing hippocampus. A recent study suggested a role of PKN1 in development of cerebellar circuits[24]. In this study we identify PKN1 as an important promoter of synapse maturation through upregulation of neuronal glutamate transporters and inhibition of mGluR-dependent synapse silencing in the hippocampus.

## Methods

Animal use and all experimental procedures were approved by the Ethical Committee for Animal Experiments of Gunma University Graduate School of Medicine and Institutional Animal Care, the Animal Care and Use Committee of Saga University (#30-007-0, 30-008-0), and Use Committee of Kobe University. All experiments were performed in accordance with the guidelines of these committees.

**Generation of PKN1a KO mice.** A genomic fragment of the mouse PKN1 gene was isolated from a 129Sv/J phage library and probed with a mouse PKN1 cDNA. For PKN1a disruption, a replacement-type targeting vector was prepared (Supplementary Fig. 1b). It contained an ~1.6 kbp HindIII–NaeI DNA fragment including the 5′ part of exon 1a, neomycin selection cassette, and an ~4 kbp XbaI/SmaI DNA fragment including exon 1b and the 5′ part of exon 2 followed by the diphtheria toxin (DT) gene for negative selection. 129 Sv/J embryonic stem (ES) cells were transfected with the linearized targeting vector by electroporation. One of the 192 G418-resistant ES cell clones carried a correctly targeted PKN1a allele, as assessed by PCR analysis and Southern blot analysis of genomic DNA.

The ES clone was injected into C57BL/6J blastocysts to generate chimeric mice. High-percentage male chimeric mice, as determined from the agouti coat color, were mated with C57BL/6J mice to determine germline transmission. F1 mice were backcrossed at least 12 times with mice with the C57BL/6J background before phenotypic analysis. PKN1a knockout mice carrying the homozygous deletion (PKN1a−/−) were viable, born at a frequency expected for Mendelian inheritance.

PKN1 expression in various tissues from wild type (PKN1a+/+) and knockout (PKN1a−/−) mice was analyzed by immunoblot analysis using the αC6 antibody against the C-terminal region of PKN1 (Supplementary Fig. 2a). The content of PKN1 in mutant mice was less than that in wild-type mice, but PKN1 remained to some extent depending on the tissue type. Biochemical measurement of PKN1 showed that the content in the brain of PKN1a−/− mice was reduced to ~1/10 of that of wild-type mice and the PKN1 content in the spleen was reduced to ~60% of that of wild-type mice (Supplementary Fig. 2b). To confirm that PKN1b remains in PKN1a−/− mice, an anti-PKN1b specific antibody, αN1b2, was prepared by immunizing rabbits with N-terminal 12 aa encoded by exon 1b conjugated with KLH (Supplementary Fig. 2c). The sensitivity of the αN1b2 antibody was not sufficient to detect endogenous PKN1b in crude tissue lysates. However, PKN1b was detected by αN1b2 in the immunoprecipitates obtained using αN2, a variant nonselective anti-PKN1 antibody (Supplementary Fig. 2d). The αN2 immunoprecipitates from PKN1a−/− mice contained more PKN1b detected by αN1b2 than those from wild-type mice, which was evident in the spleen. This finding suggests that PKN1b remains in PKN1a−/− mouse tissues, and is particularly abundantly in the spleen, consistent with the contents of PKN1 shown in Supplementary Fig. 2b.

As shown in Supplementary Fig. 2e, the amount of PKN2 did not significantly change even in the absence of PKN1a expression in various brain regions. PKN2 seems not to compensate, at least at the protein level, for the loss of PKN1a.

**Genotyping.** Genomic DNA was isolated from ES cells and mouse tail snips by standard techniques and subjected to Southern blot analysis and PCR analysis for identification. Southern blot analysis was performed using genomic DNA digested with SacI/BamHI and XhoI/BamHI and probed with probe A (SacI/HindIII fragment) and probe B (SmaI/SacI fragment), respectively. Wild type (+) and mutant alleles (−) were indicated by the presence of a 2.8 kbp (+) versus 5.8 kbp (−) SacI/BamHI for probing with probe A, and a 8.7 kbp (+) versus 11.7 kbp (−) XhoI/BamHI DNA fragment for probing with probe B (Supplementary Fig. 1c).

**Antibody.** The polyclonal antibodies αN2 and αC6 were prepared by immunizing rabbits with the fragments of N-terminal 390 aa of rat PKN1a and C-terminal 84 aa of rat PKN1 synthesized in bacteria, respectively[60]. The polyclonal antibody αN1b2 was raised by immunizing rabbits with the N-terminal 12 aa of PKN1b conjugated with KLH. The αParN2 antibody was raised by immunizing rabbits with the N-terminal 506 aa of human PKN2 synthesized in bacteria. The monoclonal anit-PKN1 and polyclonal anti-EAAT3 antibodies were purchased from BD Transduction Laboratories and Cell Signaling Technology (#12179), respectively.

**Immunoblot analysis.** Samples were subjected to SDS-PAGE, and separated products were subsequently transferred to a polyvinylidene difluoride membrane. The membrane was then blocked with phosphate-buffered saline (PBS) (20 mM sodium phosphate, pH 7.5, 137 mM NaCl) containing 0.05% Triton X-100 (PBST) and 5% normal goat serum for 1 h at room temperature. The membrane was then incubated in PBST and the primary antibody for 1 h at room temperature. The membrane was washed three times (5 min each time) in PBST before incubating the blot in PBST containing the secondary antibody conjugated to horseradish peroxidase at 1:2000 dilution for 45 min. After this incubation, the membrane was subjected to three 10 min washes in PBST. Blots were developed by the enhanced chemiluminescence method.

**Quantification of PKN1.** The contents of PKN1 in various tissues were determined by immunochemical analysis using the αC6 antibody as described in ref. [61]. Various amounts (1–10 ng) of the purified GST-fused C-terminal 310 aa region of mouse PKN1 were used as standards. Mouse tissues were removed quickly after decapitation and added to 10 vol of 50 mM Tris/HCl, pH 7.5, containing 5 mM EDTA, 5 mM EGTA, 0.5 mM dithiothreitol (DTT), 10 μg/ml leupeptin, and 1 mM phenylmethyl sulfonylfluoride (PMSF). The tissues were homogenized with 10 strokes of a Teflon/glass homogenizer and the crude lysates were used for quantification of PKN1.

**Immunoprecipitation experiment.** The brain and spleen from wild type and PKN1a−/− mice were homogenized with a polytron in 19 vol of ice-cold buffer A containing 50 mM Tris-HCl, pH 7.5, 1 mM EDTA, 1 mM EGTA, 1 μg/ml leupeptin, 1 mM DTT, 1 mM PMSF, 0.5% Triton X-100, and 150 mM NaCl. After centrifugation at $100,000 \times g$ for 15 min, the supernatant was transferred to new tubes and incubated with 3 μl of αN2 serum at 4 °C for 2 h with rotation. Eighty microliters of 50% slurry of protein A Sepharose preequilibrated with buffer A was added to the mixture and incubated for 1 h. After centrifugation at $5000 \times g$ for 1 min, the resulting pellet was washed three times with buffer A, and immuno-precipitated proteins were eluted with SDS sample buffer. Eluates were subjected to immunoblotting with the αPKN1 monoclonal antibody or αN1b2 antibody.

**Immunohistochemistry.** Wild-type mice were deeply anesthetized with sodium pentobarbital (25 mg/kg, i.p.). They were perfused transcardially with 0.1 M PBS (pH 7.4) followed by a fixative consisting of 2% paraformaldehyde and 0.2% parabenzoquinone in 0.1 M phosphate buffer (PB, pH 7.4). The brains were removed, postfixed in the same fixative for 3 h, transferred into 20% sucrose in 0.01 M PBS (pH 7.4) for 24 h, and cut into 20-μm-thick coronal sections in a cryostat. The sections were pretreated with 0.3% $H_2O_2$ for 30 min to block endogenous peroxidase activity and incubated with rabbit anti-PKN (0.1 μg/ml) IgG in 0.1 M PBS containing 0.3% Triton X-100 (0.1 M PBST) for 48 h at 4 °C. The sections were incubated with biotinylated anti-rabbit IgG (1: 200) for 1 h, and then an avidin-biotin-peroxidase complex solution (Vectastain ABC Kit, Vector Labs., 1:400) for 1 h. After each incubation, the sections were washed for 15 min with 0.1 M PBST and the immunostaining was visualized with a solution containing 0.01% 3,3″-diaminobenzidine tetrahydrochloride (DAB), 0.6% nickel ammonium sulfate, and 0.00015% $H_2O_2$.

**Electron microscopy.** PKN1a KO and wild-type mice were intravascularly perfused with a fixing solution containing 2.5% glutaraldehyde and 2.0% paraformaldehyde in 0.1 M PB (pH 7.4). After perfusion, tissue blocks of the hippocampi were extracted and immersed in the same fixative, and washed with 7.5% sucrose for 5 min. After postfixation with 1.0% OsO₄ for 2 h, the small blocks of specimens were dehydrated and embedded in an Epon-812 mixture. Ultrathin sections were cut using an ultramicrotome (LKB Ultratome® NOVA). The sections were stained with uranyl acetate and lead citrate, and then the stratum radiatum of the CA1 region approximately 50–100 μm apart from cell layer was observed under a transmission electron microscope (JEOL JEM-1010) at an accelerating voltage of 80 kV. The length of PSD was measured using Object J (https://sils.fnwi.uva.nl/bcb/objectj/index.html), a plug-in for Image J (https://imagej.nih.gov/ij/index.html).

**Golgi staining**. Rapid Golgi staining was performed using the FD Rapid GolgiStain Kit (FD NeuroTechnologies) as previously described[62]. Mice were killed by decapitation under deep isoflurane anesthesia and the removed brains were washed in ice-cold PB. Whole brains were treated for silver impregnation for 2 weeks, cryoprotected for 1 week, and sectioned at 100 μm on a cryostat. After sectioning, sections were developed, clarified, then coverslipped in resinous medium. During staining, image acquisition, and analysis, observers were blind to the genotype of each animal. Using an upright microscope (Axioplan, Carl Zeiss) with a ×40 0.75 numerical aperture objective lens and a cooled CCD camera (CoolSnap fx, Photometrics), we obtained z-stack images (0.5 μm interval) of a primary apical dendritic segment (approximately 50–100 μm from the soma) of a CA1 pyramidal neuron in the hippocampus. We analyzed 30 dendritic segments from three KO mice or three wild-type littermates blind to genotype, and measured spine head width, spine length, and spine density per 1 μm dendrite using Object J. Dendritic protrusions were classified into filopodia (processes longer than 1.0 μm without a head), stubby (shorter than 1.0 μm without a head or with a head whose diameter was smaller than half of its length), thin (longer than 1.0 μm with a head whose diameter was smaller than half of the length), and mushroom (with a head whose diameter was larger than half of its length) spines[26,27].

**Slice electrophysiology**. Synaptic transmission was recorded from mouse hippocampal slices as described previously[6,11,13]. Slices were prepared from postnatal 7- to 18-day-old (P7–18) PKN1a KO or their wild-type littermates in ice-cold oxygenated (95% $O_2$/5% $CO_2$) artificial cerebrospinal fluid (ACSF), consisting of (in mM) 119 NaCl, 2.5 KCl, 26.2 NaHCO₃, 1 NaH₂PO₄, 4 CaCl₂, 4 MgSO₄, and 11 glucose (pH 7.4). Slices were incubated in a submersion-type incubation chamber for at least 2 h at room temperature, and then transferred to a recording chamber mounted on an upright microscope (BX51WI, Olympus) equipped with IR-DIC optics. Slices were perfused with the oxygenated (95% $O_2$/5% $CO_2$) ACSF that contained 100 μM picrotoxin at ~32 °C. We recorded fEPSPs using a patch pipette that had a broken tip and was filled with normal ACSF from the CA1 hippocampal region using a Multiclamp 700A (Molecular Devices). Acquisition and analysis were performed using custom Igor Pro (WaveMetrics) software routines. Two separate Schaffer collateral/commissural pathways were stimulated using two glass electrodes placed in the *stratum radiatum* on each side of the recording electrodes (two-pathway experiments). When we performed one-pathway experiments (e.g., for LTP and PPF), we put two hippocampal slices in a recording chamber, placed one recording electrode and one stimulating electrode in a slice, and recorded fEPSPs from two slices simultaneously.

EPSPs were recorded from CA1 pyramidal cells in the current-clamp mode with a Multiclamp 700A. Recording electrodes were filled with an internal solution containing (in mM) 135 K gluconate, 10 HEPES, 0.2 EGTA, 10 KCl, 4 Mg-ATP, and 0.5 Na₃GTP (pH 7.2 with KOH, osmolarity adjusted to 275–285 mOsm). Basal synaptic transmissions were obtained at 0.05 Hz. We applied LFS (1 Hz for 15 min) only to one pathway, induced homosynaptic LTD of fEPSPs or EPSPs, and observed whether heterosynaptic LTD was induced in another pathway. The independence of two pathways was confirmed as follows: single stimuli to one and the other pathways were applied 100 ms apart, and we observed the lack of cross-facilitation between the two pathways.

When AMPAR/NMDAR, mEPSCs, NMDAR-EPSCs, or silent synapses were assessed, EPSCs were recorded in the voltage-clamp mode through a glass recording pipette with an internal solution containing (in mM) 135 Cs-MeSO₄, 10 HEPES, 0.2 EGTA, 8 NaCl, 4 Mg-ATP, and 0.3 Na₃GTP (pH 7.2 with CsOH, osmolarity adjusted to 275–285 mOsm). AMPAR-EPSCs were recorded at −70 mV and NMDAR-EPSCs were recorded at +40 mV in the presence of 10 μM CNQX. In silent synapse experiments, small AMPAR-EPSCs were evoked at −70 mV at 0.1 Hz through minimal stimulation at a reduced intensity at which some failures were surely identified visually, and then NMDAR-EPSCs were recorded using the same stimuli at +40 mV. Initially, failure rates at −70 mV ($F_{-70}$) and +40 mV ($F_{-70}$) were estimated by doubling that of events with an amplitude of more than and less than zero, respectively[63,64]. The percentage of silent synapses was calculated using the formula $1 - \ln (F_{-70})/\ln (F_{+40})$[63,64]. In these minimal stimulation experiments, LTD was induced by 1 Hz 5 min afferent stimulation paired with −50 mV 150 ms pulses. mEPSCs were recorded at 2 kHz in the presence of TTX (1 μM) and analyzed with Mini Analysis software (Synaptosoft) blind to genotype. The threshold mEPSC amplitude was set at 5 pA and each mEPSC was initially detected automatically and verified visually. NMDAR-EPSCs were recorded at −70 mV in low-$Mg^{2+}$ ACSF containing (in mM) 119 NaCl, 2.5 KCl, 26.2 NaHCO₃, 1 NaH₂PO₄, 4 CaCl₂, 0.1 MgSO₄, 11 glucose, 0.1 picrotoxin, and 0.01 NBQX[31]. TBOA was loaded through a patch pipette, whose tip was filled with a TBOA-free internal solution with 0.2% DMSO and backfilled with an internal solution with 400 μM DL-TBOA in 0.2% DMSO. In all, 10–90% rise time and decay time constant of averaged NMDAR-EPSCs were measured using Stimfit ver. 0.13 (https://github.com/neurodroid/stimfit/wiki/Stimfit)[65]. Decays were fitted to a double exponential: $I(t) = A\exp(-t/\tau f) + B\exp(-t/\tau s)$, where $I$ is the amplitude of NMDAR-EPSCs, $A$ and $B$ are the peak amplitudes of fast and slow components, and $\tau f$ and $\tau s$ are the decay time constants, respectively, and the weighted time constant ($\tau w$) was calculated as $\tau w = (A \times \tau f + B \times \tau s)/(A + B)$[11].

Drugs used were as follows: D-APV, LY341495, MPEP, MTEP, LY367385, YM298198, DL-TBOA, and (s)-DHPG (Tocris). PRL, DNER, and dynamin

inhibitory peptide and its scrambled peptide were custom-synthesized by BEX Co., Ltd. Other chemicals were from Sigma-Aldrich or Wako Pure Chemical.

**Glutamate uptake assay**. SH-SY5Y cells derived from human neuroblastoma were transfected with the expression plasmid for FLAG-tagged mouse EAAT3 (pTB701/EAAT3) using NEPA21, and were seeded within a 24-well plate at approximately 40,000 cells/well. After 24 h the cells were transfected with the expression plasmids for FLAG-tagged full-length PKN1a (pRc/CMV/PKN1a-FL), the kinase-negative mutant (pRc/CMV/PKN1a-T774A-FL), or mock using FuGENE HD transfection reagent (Roche). [³H]-Glu uptake assay was carried out 72 h after the EAAT3 transfection[66]. In brief, the culture medium was removed and replaced with Krebs-Ringer-HEPES (KRH) buffer containing (in mM) 120 NaCl, 4.7 KCl, 2.2 CaCl₂, 25 HEPES, 1.2 MgSO₄, 1.2 KH₂PO₄, and 10 glucose (pH 7.4). After a 15-min pre-incubation at 37 °C, the cells were incubated for an additional 15 min in the presence of 100 nM [³H]-Glu. The uptake of [³H]-Glu was stopped by three washes with cold KRH buffer containing 1 mM DL-TBOA, and the cells were then treated with RIPA buffer containing (in mM) 25 HEPES, 0.5% Triton X-100, 100 NaCl, and 2 EDTA (pH 8.0). The radioactivity of cell extracts diluted in scintillation cocktail (Clear-Sol II, Nakalai Tesque) was measured with a liquid scintillation counter (Beckman). The obtained results were used as the total uptake of the cells. The [³H]-Glu uptake level in the presence of 1 mM DL-TBOA was also measured and used as the nonspecific glutamate uptake level of the cells. The specific glutamate uptake level was obtained by subtracting the nonspecific glutamate uptake level from the total uptake level. To calculate the uptake level per mg protein of the cells, after counting the [³H]-Glu uptake, the protein concentration was measured from the rest of the cell extracts using the BCA Protein Assay Reagent (Thermo Scientific).

**Statistics and reproducibility**. Results are reported as mean ± s.e.m. For multiple statistical comparisons, one-way ANOVA with Tukey's post hoc test in EZR (easy R)[67] was used, except when Holm's test was adopted for multiple comparisons with repeated measures (Fig. 8d). Comparisons between two groups were performed using unpaired, two-tailed t-test with Welch's correction (Welch's t-test) or paired t-test in Excel. Two groups that do not follow a normal distribution were compared using Mann–Whitney's U test in Excel (Supplementary Fig. 4). We did not statistically calculate sample sizes; however, the numbers of data we got in experiments presented here were similar to those in papers previously published in our field. We usually used at least three mice for one group in each experiment.

**Reporting summary**. Further information on research design is available in the Nature Research Reporting Summary linked to this article.

## Data availability
The data that support the findings of this study are readily available from the corresponding authors upon reasonable request. Source data are available in Supplementary Data 1.

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

## Acknowledgements

We thank Drs. R. Malenka, J. Isaac, and A. Terashima for valuable comments. This work was funded by grants-in aid for scientific research Nos. 11005815, 09152356, and

08093928 to H. Yasuda. and 18022025 to H. Mukai from the Ministry of Education, Culture, Sports, Science and Technology, a grant from Takeda Science Foundation to H. Yasuda, and a joint research program of Biosignal Research Center, Kobe University.

## Author contributions

H. Yasuda and H. Mukai designed this research; H. Yamamoto and H. Mukai performed glutamate uptake assay; K.H., T.K., T.S., and H. Yasuda performed histological analyses; H. Morisaki, M. Miyamoto, S.T., Y.O., and H. Mukai generated PKN1a KO mice; H. Yasuda performed electrophysiological experiments and analyses; H. Yasuda, M. Mehruba, and H. Mukai performed biochemical analyses; and all authors wrote the paper.

## Competing interests

The authors declare no competing interests.
