## [Peer Review File · Communications Biology]

Reviewers' comments:

Reviewer #1 (Remarks to the Author):

This is an excellent manuscript, in which the authors provided compelling results demonstrating a potential molecular mechanism through which heterosynaptic plasticity is inhibited in normal physiological conditions. Namely, the authors identified that PKN1a, through regulating glutamate transporters, confined glutamate spreading and activation pattern of mGluRs, limited LTD within the pathway that received LTD induction, while this pathway-specific induction was abolished in PKN1a KO mice. The experiments are well designed, electrophysiological data are elegant, and conclusions are reasonable. Overall, it is an outstanding study, with only minor concerns.

Below are a few points for the authors to consider:

1) Figure titles: May the authors consider using the summaries of the experimental results rather than the authors' conclusions based on the results for figure titles? For example, the title of figure 1 reads "PKN1 promotes maturation of synapses." Would the authors consider simply stating that "AMPA-mediated synaptic transmission is weakened in PKN1 KO mice?"

2) The spine results show an overall reduction of the spine density in PKN1 KO. This result does not directly support the authors' conclusion that maturation of synapses is affected. Rather, this result suggests an overall reduction in synaptic connectivity. It is possible that a portion of the thin or immature spines were not detectable in KO samples. In this case, comparing the subcategories of spines may help substantially.

3) The electrophysiology about homo- vs. heterosynaptic plasticity is beautifully delivered. However, the patterns of LTD are somewhat different among experiments. For example, Fig. in 3a right WT TBOA summary, the EPSP amplitudes kept decaying after LTD induction, a pattern of reduction very different from previous summaries. A similar "rundown" like phenomenon was observed in Fig.4b right. Any explanations?

4) Fig. 5: the failure rates of EPSCs at -70 mV and +40 mV were sampled and compared separately, with the results suggestive of generation of silent synapses. A more direct comparison incorporating the failure rates at both the holding potentials may be worth considering (e.g., equations used in the Liao 1995 Nature) to depict a better view of % silent synapses, similar to what the authors used in Fig. 6.

Reviewer #2 (Remarks to the Author):

In this study Yasuda et al. show the role of PKN1, a serine/threonine protein kinase, in long-term depression and synapse maturation in the hippocampus using a novel PKN1 KO mouse model.

The topic is interesting. Recently published results have shown that PKN1 contributes to synapse maturation in the cerebellum (zur Nedden et al JCI 128:2076, 2018), but as the authors correctly point out the physiological roles of PKN1 in the brain remain unclear. Furthermore, the mechanisms that convert silent synapses into functional synapses in the immature brain are not understood, and the findings that PKN1 prevents mGluR LTD and "unsilences" synapses in the immature hippocampus constitute important results.

The manuscript begins with a description of a novel PKN1a KO mouse model. The authors provide experimental evidence that in this mouse model, a well-established LTD protocol that normally induces only NMDAR-dependent LTP causes mGluR5-dependent LTD. Next they show that TBOA, a glutamate transporter antagonist, recapitulates the mGluR-dependent LTD (but see major point 2

below). Importantly, mGluR5-dependent LTD in PKN1a KO mice is occluded in the presence of TBOA. Dominant negative PKN1 constructs are used to demonstrate that PKN1 activates EAAT3, a neuronal glutamate transporter, in SH-SY5Y cells (see major point 3 below). The authors then characterize LTD in PKN1a KO mice using whole cell recordings and show that PKN1 suppresses LTD mediated by endocytosis at postsynaptic sites. They demonstrate, using a minimal stimulation protocol, that LTD in PKN1 KO mice is associated with an increase in silent synapses, while this is not the case in WT mice. Finally they show that mice lacking PKN1 have a higher percentage of silent synapses among naive hippocampal synapses than WT mice.

The experiments appear to be technically sound and well analyzed. With the exception of a few points which the authors should address, overall the data convincingly support the claim that "PKN1 critically promotes synapse maturation by inhibiting mGluR-dependent silencing through neuronal glutamate transporter activation".

Major points

1. A recent article showed that Pkn1^{-/-} mice have a defective parallel fiber-Purkinje cell synapse formation (zur Nedden et al JCI 128:2076, 2018) and should be briefly discussed in the introduction (it is alluded to at the end of the discussion).
2. TBOA-induced LTD is blocked with a co-application of MPEP and LY367385, which does not allow to assess the specific effect of TBOA on mGluR5-dependent LTD. This is important since the underlying hypothesis is that TBOA recapitulates the "unmasked" LTD in PKN1 KO mice, which is strictly mGluR5-dependent. What is the effect of the LFS protocol in TBOA + LY367385? Does it still elicit mGluR5 LTD?
3. A weakness is that while the role of PKN1 on glutamate uptake is demonstrated in SH-SY5Y cells, a proper demonstration that EAAT3 function is affected in PKN1 KO neurons is lacking. I understand that EAAT3-mediated currents are too small to be resolved in neurons. Nevertheless, an indirect but relatively simple assessment of glutamate clearance would be to measure NMDAR currents at -70mV in 0.1 mM Mg²⁺ ACSF (see Li et al. Neuron 62:788, Figure 4) in WT and PKN1^{-/-} mice. If glutamate clearance is affected, as the authors suggest, then neurotransmitter spillover may be enhanced leading to increased diffusion distance and activation of extrasynaptic NMDARs, hence increased rise time and decay time of NMDAR currents.

Minor points

1. Page 4 line 65. The view that mGluR5 antagonists may treat Fragile X is not in line with the most recent scientific literature. While mGluR5 antagonists were promising in animal models, they failed in clinical trials. See for instance Erickson et al., J Neurodev Disord. 9: 7 (2017).
2. Page 9 line 159. The rationale for examining glutamate clearance after finding out that mGluR LTD is affected in PKN1^{-/-} mice is not clear. Adding a reference to previous work showing that mGluR-LTD can be modulated by glutamate clearance might help (e.g. Li, S. et al Neuron 62:788).
3. Page 17 line 288, I would suggest: "There are few reports on heterosynaptic LTD and..."
4. Page 18 lines 307-310: given the upregulation of NMDARs as well as AMPARs in Tsc1-lacking neurons, and the numerous pathways affected by knocking out Tsc1, I do not find the argument convincing. The sentence should be deleted.
5. page 18 line 311: has
6. page 19 line 317: "EAAT3 may be" instead of "EAAT3 is".
7. page 19 line 320: " we identify PKN1 as an important promoter..."

Reviewer #3 (Remarks to the Author):

In this paper, the authors generated and analyzed synaptic properties of PKN1a KO mice. They found that these mice have reduced spine density and size, reduced basal synaptic strength, normal LTP, enhanced LFS-LTD, reduced AMPA receptor current and more silent synapses. The enhanced LTD is mGluR5 dependent and can be mimicked by the glutamate transporter inhibitors.

Taken together, the authors concluded that PKN1a upregulates the glutamate transporter to suppress mGluR5-dependent LTD and promote synapse maturation.

In general, this is an important study. The most interesting aspect of the paper is the enhanced mGluR5-LTD in the KO mice, which is normally absent in developing hippocampus in WT. The connection to the glutamate transporters and silent synapses is also interesting, providing a potential mechanism for developmental regulation of synapse maturation. I have a number of comments and suggestions that might be helpful to improve the paper.

General comments:

- (1). LTD was induced by LFS in the study. However, LTD in the KO mice was partially mGluR5-dependent. To directly test this form of LTD, PP-LFS in the presence of APV should be used to verify this conclusion.
- (2). The authors concluded that PKN1a regulates mGluR-LTD via upregulating the glutamate transporter. While this conclusion appears to be consistent with the transporter inhibitor experiments, the overall evidence for this is very weak. To the minimum, the authors should test (a) is any of the transporters altered in the KO mice (protein level, surface expression, post-translation modifications etc) and (b) are the phenotypes of KO mice be rescued by reintroducing the transporters or pharmacological manipulations.
- (3). The enhanced mGluR-LTD in the KO mice could also be explained by changes in spine morphology (immature spines) and/or mGluR5 distribution. Because the KO mice have smaller spines/PSD, it is possible that the expression of mGluR5 in the KO mice is physically closer to the presynaptic active zone, therefore it can be activated by LFS more easily than in WT mice, thus the presence of mGluR5-LTD in the KO mice. This hypothesis is also consistent with the results obtained with the transporter inhibitor.

Specific comments:

- (1). The age of the mice should be indicated in each experiment.
- (2). Supplementary Fig 2: It is important to quantify the level of PKN1b in WT and KO to determine whether 1b level is altered. In f and g, staining images for the KO mice should also be provided.
- (3) Fig 1. The quality of the Golgi staining is not sufficient. It's hard to resolve the spine morphology, or even density.
- (4). Fig 2. LTP by HFS was not altered in the KO mice. What about LTP induced by other protocols (e.g., TBS)? What about late phase LTP? These are important questions to know because the authors appeared to rule out the possibility that LTP deficits may contribute to the silent synapse phenotype in the KO mice.
- (5). Page 8 last sentence: I am not sure if this statement is accurate. My understanding is that most previous studies support that mGluR5-LTD in the hippocampus is protein synthesis dependent.
- (6). Fig. 3. As indicated above in the general comments, it is important to know how PKN1a affects glutamate transporters (protein level, surface expression etc). Is it possible to measure the endogenous glutamate level?
- (7). Fig. 4. I am not clear why PRL was tested only in WT, but not in KO, and the Dyn inhibitor tested only KO, but not in WT mice?

(8). Fig 5 and 6. The failure rates under basal conditions in Fig 5 was the same between WT and KO mice, then why they were different in Fig. 6 between genotypes? Was the age of the mice in these two experiments different? More silent synapses in the KO mice suggest reduced synaptic AMPA receptors. Some biochemical/staining experiments would be helpful to confirm this.

(9). Fig 7. Model. Should the KO spine smaller??

Answers to comments from Reviewer #1:

1) Figure titles: May the authors consider using the summaries of the experimental results rather than the authors' conclusions based on the results for figure titles? For example, the title of figure 1 reads "PKN1 promotes maturation of synapses." Would the authors consider simply stating that "AMPA-mediated synaptic transmission is weakened in PKN1 KO mice?"

[Answer]

We agree with the comment of reviewer 1 that the figure titles should reflect experimental results. We have changed figure titles as follows.

Page 45, line 817; "Figure 1: PKN1 promotes maturation of synapses." -> "Figure 1: AMPAR-mediated synaptic transmission is weakened in PKN1a KO mice."

Page 47, lines 830–831; "(new) Figure 2: PKN1a KO mice have smaller spines with shorter PSDs in the hippocampus than wild-type mice."

Page 49, lines 842–page 50, line 843; "(original) Figure 2: PKN1 masks mGluR5- and translation-dependent spreading LTD." -> "Figure 3: Heterosynaptic LTD is induced by single-pulse LFS in PKN1a KO mice and blocked by an mGluR5 inhibitor."

Page 51, lines, 859–860; "(new) Figure 4: Heterosynaptic LTD induced by paired-pulse LFS is inhibited by simultaneous application of mGluR1 and mGluR5 antagonists in PKN1a KO mice."

Page 54, lines 883–884; "(original) Figure 3: mGluR-LTD in PKN1a KO mice is induced through impaired neuronal glutamate transporter activity."-> "Figure 5: Heterosynaptic LTD is induced in wild-type hippocampus in the presence of glutamate transporter inhibitor and blocked by mGluR1 and mGluR5 inhibitors."

Page 56, line 916; "(new) Figure 6: Decay of NMDAR-ESPCs is prolonged in PKN1a KO and TBOA-loaded neurons."

Page 59, lines 929–930; "(original) Figure 4: PKN1a suppresses mGluR-LTD at postsynaptic sites" -> "Figure 7: Inhibition of PKN1 and endocytosis in postsynaptic neurons unmasks homo- and heterosynaptic LTD"

Page 62, line 969; "(original) Figure 6: PKN1 reduces silent synapses in normal developing hippocampus" -> "Figure 9: PKN1 KO mice possesses more silent synapses than wild-type mice."

Supplementary information

Page 7, line 58; "Supplementary Figure 5: PKN1 does not affect LTP." -> "Supplementary Figure 5: Similar LTP is induced in both wild-type and PKN1a KO mice."

Page 8, lines 68–69; "Supplementary Figure 6: Additional LTD abnormally induced in PKN1a KO mice is mediated by group 1 mGluRs and independent of NMDA receptors and intracellular calcium." -> "Supplementary Figure 6: Additional LTD abnormally induced in

PKN1a KO mice is inhibited by group 1 mGluR antagonist but not by intracellular calcium chelator.”

2) The spine results show an overall reduction of the spine density in PKN1 KO. This result does not directly support the authors' conclusion that maturation of synapses is affected. Rather, this result suggests an overall reduction in synaptic connectivity. It is possible that a portion of the thin or immature spines were not detectable in KO samples. In this case, comparing the subcategories of spines may help substantially.

[Answer]

In accordance with the suggestion of reviewer 1 we have classified spines into filopodia (processes longer than 1.0 μm without a head), stubby (shorter than 1.0 μm without a head or with a head whose diameter was smaller than half of its length), thin (longer than 1.0 μm with a head whose diameter was smaller than half of its length), and mushroom (with a head whose diameter was larger than half of its length) spines, and we analyzed changes in the numbers of spine subtypes. This classification has been described in Methods (page 33, lines 564–567). We found that filopodia and mushroom spines were reduced in PKN1a KO mice. We presented these results in Fig. 2c and described them in Results (page 7, lines 115–121). Because group 1 mGluRs preferentially shrink large spines (Oh, Hill, Zito, PNAS, 2012), we discussed the possibility that the enhanced mGluR-LTD mechanisms may reduce mushroom spines in PKN1a KO mice (page 24, lines 403–405). However, the numbers of stubby and thin spines did not decrease in KO mice, indicating that knocking out PKN1a does not necessarily disturb general morphological development of spines. Therefore, we deleted the sentence “therefore, PKN1 prevents developing synapses from staying immature by suppressing mGluR5- and translation-dependent LTD” in page 18, lines 306–307 in the original manuscript (page 24, line 403 in the present manuscript).

3) The electrophysiology about homo- vs. heterosynaptic plasticity is beautifully delivered. However, the patterns of LTD are somewhat different among experiments. For example, Fig. in 3a (=Fig. 5a in the revised manuscript) right WT TBOA summary, the EPSP amplitudes kept decaying after LTD induction, a pattern of reduction very different from previous summaries. A similar “rundown” like phenomenon was observed in Fig.4b right (=Fig. 7b in the revised manuscript). Any explanations?

[Answer]

One of the reasons why heterosynaptic LTD in TBOA in previous Fig. 3a (= Fig. 5a in the revised manuscript) looks like a rundown of fEPSPs is that the Y-axis range of heterosynaptic pathway in previous Fig. 3a was 60–130%, which is narrower than that of homosynaptic pathway (30–130%), for example. Therefore, for clarity, we have changed the

range of heterosynaptic pathway in new Fig. 5a to 40–125% to be the same as that of homosynaptic pathway in Fig. 5a.

mGluR5 was found to be involved in heterosynaptic LTD induced by single-pulse LFS in KO mice (Fig. 3). On the other hand, mGluR1 and mGluR5 were involved in heterosynaptic LTD induced by single-pulse LFS in wild-type mice in the presence of 10 μ M TBOA (Fig. 5a–d). We speculate that the glutamate concentration in the synaptic cleft during single-pulse LFS is higher, and heterosynaptic LTD is induced more strongly in wild-type mice in the presence of TBOA than in KO mice. Therefore, it may take a longer time until the fEPSP amplitude becomes stable after heterosynaptic LTD induction, as shown in Fig. 5a, b.

Heterosynaptic LTD in KO mice was more robust in whole-cell recordings (Fig. 7b) than in fEPSP recordings (Fig. 3b). It could be because some washout effect may help induce more robust heterosynaptic LTD in whole-cell slice patch experiments and it may take time until the EPSP amplitude became stable after heterosynaptic LTD induction.

4) Fig. 5 (= Fig. 8 in the present manuscript): the failure rates of EPSCs at -70 mV and +40 mV were sampled and compared separately, with the results suggestive of generation of silent synapses. A more direct comparison incorporating the failure rates at both the holding potentials may be worth considering (e.g., equations used in the Liao 1995 Nature) to depict a better view of % silent synapses, similar to what the authors used in Fig. 6 (= Fig. 9 in the present manuscript).

[Answer]

We have calculated the percentages of silent synapses before and after pairing protocol in Fig. 8 experiments.

	Baseline,	30 min after LFS pairing
wild-type, LFS path	15.6 \pm 5.6%,	26.3 \pm 5.0%
wild-type, cont path	20.4 \pm 8.7%	10.6 \pm 5.2%
KO, LFS path	31.9 \pm 6.2%	60.0 \pm 3.5%
KO, cont path	28.6 \pm 5.8%	50.1 \pm 6.9%

In these experiments, NMDAR-EPSCs were recorded only once, at least 50 min after the start of whole-cell recordings, because NMDAR-EPSC recordings at +40 mV before LTD induction could induce some plastic changes in synaptic transmission especially in the developing hippocampus, which could affect the subsequent LTD experiments.

On the other hand, the percentages of silent synapses were 28.1 \pm 5.4% in wild-type mice and 67.6 \pm 6.0% in KO mice (new Fig. 9). In these experiments, we usually

started to record NMDAR-EPSCs within 15 min after break-in. Rundown of NMDAR-EPSCs could happen during prolonged whole-cell recordings (Macdonald et al., J. Physiol., 1989; Rosenmund and Westbrook, J. Physiol., 1993, etc); therefore, we are afraid that the failure rate of NMDAR-EPSCs in Fig. 8d may be underestimated and this is why the calculated percentage of silent synapses ($1 - \ln(F_{-70}) / \ln(F_{+40})$) before pairing protocol is smaller than those in Fig. 9. To avoid confusing the readers, we did not show percentage of silent synapses in Fig. 8.

Answers to comments from Reviewer #2:

Major points

1. A recent article showed that Pkn1^{-/-} mice have a defective parallel fiber-Purkinje cell synapse formation (zur Nedden et al JCI 128:2076, 2018) and should be briefly discussed in the introduction (it is alluded to at the end of the discussion).

[Answer]

We agreed with the suggestion of reviewer 2 and we have added the following sentence in the introduction: Very recently, PKN1 has been reported to be involved in axonal outgrowth and presynaptic differentiation of parallel fibers of cerebellar granule cells²⁴ (page 4 lines 73–75).

2. TBOA-induced LTD is blocked with a co-application of MPEP and LY367385, which does not allow to assess the specific effect of TBOA on mGluR5-dependent LTD. This is important since the underlying hypothesis is that TBOA recapitulates the “unmasked” LTD in PKN1 KO mice, which is strictly mGluR5-dependent. What is the effect of the LFS protocol in TBOA + LY367385? Does it still elicit mGluR5 LTD?

[Answer]

In accordance with the suggestion, we have carried out additional experiments at Saga University, and we have described on page 26, lines 440–441 stated that animal experiments were approved by the ethics committee of Saga University. LY367385 did not affect homosynaptic LTD; however, it significantly reduced heterosynaptic LTD in the presence of TBOA in wild-type mice. We have added these data on these findings in Fig. 5b, d and described them in the text (page 12, line 213–page 13, line 221). mGluR1 was also found to be involved in both homo- and heterosynaptic LTD induced by paired-pulse LTD (see Fig. 4); therefore, we agree with the involvement of mGluR1 as well as mGluR5 in mGluR-LTD in PKN1a KO mice. We have changed “mGluR5” to “group 1 mGluR” on page 5, line 84, page 11, line 195, page 21, lines 351 and 353, and page 23, line 399.

3. A weakness is that while the role of PKN1 on glutamate uptake is demonstrated in SH-SY5Y cells, a proper demonstration that EAAT3 function is affected in PKN1 KO neurons is lacking. I understand that EAAT3-mediated currents are too small to be resolved in neurons. Nevertheless, an indirect but relatively simple assessment of glutamate clearance would be to measure NMDAR currents at -70mV in 0.1 mM Mg²⁺ ACSF (see Li et al. Neuron 62:788, Figure 4) in WT and PKN1^{-/-} mice. If glutamate clearance is affected, as the authors suggest, then neurotransmitter spillover

may be enhanced leading to increased diffusion distance and activation of extrasynaptic NMDARs, hence increased rise time and decay time of NMDAR currents.

[Answer]

We agree with reviewer 2 and we have recorded NMDAR-EPSCs at -70 mV in a low Mg²⁺ ACSF. We observed no significant difference in rise time between wild-type and KO mice; however, decay time constant was prolonged in KO mice compared with wild-type mice. We also made sure that intracellular application of TBOA delayed decay time constant of NMDAR-EPSCs without any significant changes in rise time in wild-type mice. These results suggest that PKN1a deletion prolongs NMDAR-EPSC kinetics through a reduction in neuronal glutamate transporter activity. We presented these data in Fig. 6 and added these results in the text (page 14, line 238–page 15, line 258).

Minor points

1. Page 4 line 65. The view that mGluR5 antagonists may treat Fragile X is not in line with the most recent scientific literature. While mGluR5 antagonists were promising in animal models, they failed in clinical trials. See for instance Erickson et al., J Neurodev Disord. 9: 7 (2017).

[Answer]

In accordance with the comment of reviewer 2, we have deleted “, and mGluR5 antagonists may treat this syndrome” (page 4, line 62).

2. Page 9 line 159. The rationale for examining glutamate clearance after finding out that mGluR LTD is affected in PKN1^{-/-} mice is not clear. Adding a reference to previous work showing that mGluR-LTD can be modulated by glutamate clearance might help (e.g. Li, S. et al Neuron 62:788).

[Answer]

In accordance with the comment of reviewer 2, we have added the reference and changed the sentences as follows: “Next, we addressed the possibility that an elevated glutamate concentration promotes abnormal mGluR-LTD induction in immature PKN1a KO mice. Previously, reduced glutamate clearance was reported to enhance mGluR-LTD³².” (page 12,

lines 200–202).

3. Page 17 line 288, I would suggest: “ There are few reports on heterosynaptic LTD and...”

[Answer]

In accordance with the suggestion of reviewer 2, we have changed “Reports on heterosynaptic LTD are not so many^{41, 42}” to “There are few reports on heterosynaptic LTD^{51, 52}” (page 22, lines 384–page 23, line 385).

4. Page 18 lines 307-310: given the upregulation of NMDARs as well as AMPARs in Tsc1-lacking neurons, and the numerous pathways affected by knocking out Tsc1, I do not find the argument convincing. The sentence should be deleted.

[Answer]

In accordance with the comment of reviewer 2, we have deleted the following sentences: The findings that genetic inhibition of group 1 mGluR-dependent LTD by deleting the tuberous sclerosis complex 1 (Tsc1) gene in mice upregulates AMPAR-EPSCs⁴⁶ supports our idea, although neurons lacking Tsc1 showed a normal AMPAR/NMDAR ratio because NMDAR-EPSCs were also unexpectedly enhanced for some reason. (page 24, line 403)

5. page 18 line 311: has

[Answer]

In accordance with the comment of reviewer 2, we have changed “EAAT3 is modestly expressed in hippocampal neurons and have less capacity” to “EAAT3 is modestly expressed in hippocampal neurons and has less capacity” (page 24, line 413).

6. page 19 line 317: “ EAAT3 may be” instead of “ EAAT3 is”.

[Answer]

In accordance with the comment of reviewer 2, we have changed “EAAT3 is more important in synapses” to “EAAT3 may be more important in synapses” (page 25, lines 419–420).

7. page 19 line 320: “ we identify PKN1 as an important promoter...”

[Answer]

In accordance with the comment of reviewer 2, we have changed “we propose PKN1 as the critical promoter” to “we identify PKN1 as an important promoter” (page 25, line 421).

Answers to comments from Reviewer #3:

General comments:

(1). LTD was induced by LFS in the study. However, LTD in the KO mice was partially mGluR5-dependent. To directly test this form of LTD, PP-LFS in the presence of APV should be used to verify this conclusion.

[Answer]

We agree with the reviewer 3's comment and we have examined the effects of PP-LFS on synaptic transmission in PKN1a KO mice in the presence of D-APV. PP-LFS induced more robust homosynaptic LTD in KO mice than in wild-type mice. PP-LFS did not depress synaptic transmission at heterosynaptic pathways in wild-type mice; however, PP-LFS induced heterosynaptic LTD in KO mice. We examined the effects of group 1 mGluR antagonists on LTD induced by PP-LFS in KO mice and presented the data obtained in Fig. 4.

i) Homosynaptic LTD induced by PP-LFS in KO mice was significantly reduced by mGluR1 antagonists (LY367385 or YM298198) or mGluR5 antagonists (MPEP or MTEP). Because MPEP did not completely inhibit homosynaptic LTD, we tested MTEP, another mGluR 5 inhibitor, and found that its effects are similar to those of MPEP. We also used YM298198, another mGluR1 antagonist, and we found that YM298198 and LY367385 had similar significant effects on LTD induced by PP-LFS, indicating that mGluR1 is also involved in LTD induced by PP-LFS in KO mice. Interestingly, homosynaptic LTD induced by PP-LFS was not completely inhibited by simultaneous application of mGluR1 and mGluR5 antagonists. Huber's lab reported that LTD induced by PP-LFS is suppressed by 100 μ M LY341495, but is not completely inhibited by 20 μ M LY341495 + MPEP + LY367385 (Fig. 5F in Volk et al., J. Neurophys., 2006). 20 μ M LY341495 should effectively block group 2 and group 3 mGluRs except mGluR4, whose Ki value is 22.0 μ M (Howson and Jane, Br. J. Pharm. 2003). Therefore, mGluR4 might be involved in LTD induced by PP-LFS in KO mice.

ii) Heterosynaptic LTD induced by PP-LFS was suppressed by simultaneous application of mGluR1 and mGluR5 antagonists. However, mGluR1 or mGluR5 antagonist did not significantly reduce heterosynaptic LTD, presumably because the glutamate concentration during PP-LFS is high and activation of either of them is enough to induce heterosynaptic LTD. These results are described in Results (page 9, line 160–page 11, line 192) and Discussion (page 21, line 356–page 22, line 376). We admit that mGluR1 also influences PKN1-EAAT3 activity and we have changed our conclusions in the text (mGluR5 -> group 1 mGluRs on page 5, line 84, page 11, line 195, page 21, lines 351 and 353, and page 23, line 399). Also, we have summarized the involvement of mGluR1 and mGluR5 under various conditions we used in the present study in Table 1 (page 64), so that the readers can more easily understand our study.

(2). The authors concluded that PKN1a regulates mGluR-LTD via upregulating the glutamate transporter. While this conclusion appears to be consistent with the transporter inhibitor experiments, the overall evidence for this is very weak. To the minimum, the authors should test (a) is any of the transporters altered in the KO mice (protein level, surface expression, post-translation modifications etc)

[Answer]

In accordance with the comment of reviewer 3, we have examined the expression of EAAT3 by Western blotting and found that EAAT3 expression in the hippocampus is reduced in KO mice by approximately 20% of that in wild-type mice. We presented the data on this finding in Fig. 5e and Supplementary Fig. 8, and described these in Abstract (page 2, lines 36–37), Result (page 13, lines 226–229), and Discussion (page 20, lines 343–344).

and (b) are the phenotypes of KO mice be rescued by reintroducing the transporters or pharmacological manipulations.

[Answer]

These experiments that reviewer 3 suggested are very interesting. If we overexpress EAAT3 in the hippocampus of KO mice, EAAT3 expression will take a few weeks, and we are afraid that exogenous EAAT3 may not be enough at postnatal 1–2 weeks. Furthermore, because we did not find pharmacological tools to remove glutamate, we did not pursue to examine this idea.

(3). The enhanced mGluR-LTD in the KO mice could also be explained by changes in spine morphology (immature spines) and/or mGluR5 distribution. Because the KO mice have smaller spines/PSD, it is possible that the expression of mGluR5 in the KO mice is physically closer to the presynaptic active zone, therefore it can be activated by LFS more easily than in WT mice, thus the presence of mGluR5-LTD in the KO mice. This hypothesis is also consistent with the results obtained with the transporter inhibitor.

[Answer]

We agree that group 1 mGluRs, many of which are expressed at perisynaptic sites in the normal hippocampus, could be closer to the active zone in shorter PSDs on smaller spine,s and glutamate could activate mGluRs to a greater extent. However, we are not sure whether PKN1 directly regulates spine morphology without affecting synaptic plasticity. In this case, spine shrinkage is the cause and mGluR-LTD is the result. On the other hand, we showed that PKN1a inhibition downregulates EAAT3 expression and activity, and that inhibition of neuronal glutamate transporters mimics electrophysiological phenotypes of knocking out

PKN1a; it unmasks mGluR-LTD and prolongs NMDAR-ESPC decays. Presumably, neuronal glutamate transporters do not directly affect the neuronal cytoskeleton. Furthermore, recent literature suggests that activation of mGluR induces synaptic weakening and spine shrinkage (e.g., Oh et al., PNAS, 2012; Ramiro-Cortés et al., PLOS ONE, 2013). Therefore, we think that it is more likely that mGluR-LTD is initially induced, which causes spine shrinkage in PKN1a KO mice. We added these two references (refs. 56 and 57; page 24, line 403). However, perisynaptic mGluRs might be more activated on smaller spines with shorter PSDs in KO mice. Therefore, we described this possibility in Discussion as follows: Many mGluRs are expressed at perisynaptic sites²⁹. Because PKN1a KO mice have smaller spines (Supplementary Fig. 4) with shorter postsynaptic densities (Fig. 2d) than those in wild-type mice, mGluRs could be physically closer to presynaptic active zones. The possible closeness of group 1 mGluRs to active zones might also help induction of mGluR-LTD in PKN1a KO mice. (page 24, lines 409–412).

Specific comments:

(1). The age of the mice should be indicated in each experiment.

[Answer]

In accordance with the comment of reviewer 3, we have indicated the ages of mice in the figures and their legends.

Fig. 1a–c in page 45 and its legend on page 45, lines 820 and 822, and page 46, lines 824 and 826–827.

Fig. 2a, d on page 47 and its legend on page 47, lines 832–page 48, lines 834, and 838–839.

Fig. 3a on page 49, and its legend on page 50, lines 844 and 849.

Fig. 4 legend on page 51, line 862.

Fig. 5 legend on page 54, lines 888, 890–892, and page 55, lines 904–906.

Fig. 6a, b on page 56 and its legend on page 56, line 918, and page 57, line 922.

Fig. 7a on page 58 and its legend on page 59, lines 931–932, 934, 939, and 942.

Fig. 8a, b on page 60 and its legend on page 60, line 949, and page 61, line 950.

Fig. 9a on page 62 and its legend on page 62, lines 970–971, and 974–975.

Supplementary information

Supplementary Fig. 4 legend on page 6, line 51.

Supplementary Fig. 5 legend on page 7, lines 62, and 64.

Supplementary Fig. 6 legend on page 8, line 71, and page 9, line 78.

Supplementary Fig. 7 legend on page 9, line 90.

Supplementary Fig. 8 legend on page 10, lines 96.

Supplementary Fig. 9 legend on page 11, line 107.

(2). Supplementary Fig 2: It is important to quantify the level of PKN1b in WT and KO to determine whether 1b level is altered. In f and g, staining images for the KO mice should also be provided.

[Answer]

(PKN1b)

Unfortunately, the sensitivity of the PKN1b-specific antibody used in our study (α N1b2 antibody) was not sufficient to detect endogenous PKN1b in whole-cell lysates, as described on page 27, lines 467–468 in Methods and on page 4, lines 31–32 in supplementary information. Therefore, we were unable to measure the PKN1b content in crude lysate of each tissue. However, this PKN1b-specific antibody could detect PKN1b included in total PKN1 immunoprecipitated from brain and spleen lysates, as shown in the right lower panel of the supplementary Fig. 2d. The intensity of PKN1b immunoreactivity was higher in the PKN1 immunoprecipitates from PKN1a KO mouse samples than in those from wild-type mouse samples, suggesting that PKN1b is dominant in PKN1a KO tissues. The total PKN1 content in the PKN1a KO brain is $\sim 1/10$ of that in the wild-type brain (Supplementary Fig. 1b) even if PKN1b expression increases to compensate for the lack of PKN1a; therefore, the PKN1b content seems to be less than $1/10$ of the total PKN1 content in the wild-type brain. (KO mouse staining)

Dr. Toshio Kawamata, an expert of immunohistochemistry who previously reported the immunohistochemical localization of PKN1 in the human brain (J. Neurosci., 1998;18: 7402–7410), performed the present experiment in Supplementary Figs. 2f and 2g. Unfortunately, he already passed away and it is difficult to obtain control hippocampal images of PKN1a KO mice with the same condition and quality as those shown in Figs. 2f and 2g. The quality of the α C6 antibody used for this experiment for immunohistochemical staining was supported for mouse brain tissue (Supplementary Fig. 3a), and was also proved to be applicable to human brain tissue (J. Neurosci. 1998;18: 7402–7410). The α C6 antibody does not discriminate PKN1a from PKN1b; therefore, some PKN1 immunoreactivity (corresponding to PKN1b) remains in the brain of PKN1a KO mice, suggesting that PKN1a KO mouse tissue cannot be an ideal negative control for immunostaining using the α C6 antibody. Thus, we omitted the staining image of the PKN1a KO mouse brain in the revised manuscript.

(3) Fig 1. The quality of the Golgi staining is not sufficient. It's hard to resolve the spine morphology, or even density.

[Answer]

We analyzed spine morphology using serial images at 0.5 μm intervals of Golgi stained dendrites (see example images below). We selected well-focused spines and measured their length and width. These original images had good resolution.

Example original images of Golgi stained dendritic spines.

The images we previously presented in Fig. 1e were montages of these original images we

analyzed, and we agreed that spines on dendrites were not well resolved in the previous images, presumably because the images were too small and the background was too bright. Therefore, we regenerated and presented magnified z-stack images of Golgi stained dendrites (Fig. 2a).

(4). Fig 2. LTP by HFS was not altered in the KO mice. What about LTP induced by other protocols (e.g., TBS)? What about late phase LTP? These are important questions to know because the authors appeared to rule out the possibility that LTP deficits may contribute to the silent synapse phenotype in the KO mice.

[Answer]

In accordance with the comment of reviewer 3, we have examined the effects of TBS and 4 HFS protocols on synaptic transmission in KO mice. TBS and 4 HFS protocols induced LTP also in KO mice, and the amplitude of LTP was not different between wild-type and KO mice. These results were added in Supplementary Fig. 5 and described in the text (page 7, line 127–page 8, line 130).

(5). Page 8 last sentence: I am not sure if this statement is accurate. My understanding is that most previous studies support that mGluR5-LTD in the hippocampus is protein synthesis dependent.

[Answer]

The paper from Huber's lab we cited here (ref. 30) showed that mGluR-LTD in the P8–15 hippocampus was not affected by anisomycin or cycloheximide (Figs. 2 and 3), and they concluded that mGluR-LTD is independent of protein synthesis at these ages.

<http://www.jneurosci.org/content/25/11/2992.long>

(6). Fig. 3 (= Fig. 5 in the present manuscript). As indicated above in the general comments, it is important to know how PKN1a affects glutamate transporters (protein level, surface expression etc). Is it possible to measure the endogenous glutamate level?

[Answer]

It would be interesting to determine the glutamate concentrations during LFS in wild-type and KO mice; however, we have not measured extracellular glutamate concentration, and we were able to find any conventional methods to monitor glutamate concentration in the synaptic cleft. Instead, we examined the effects of knocking out PKN1 and intracellular loading of TBOA on NMDAR-EPSCs at -70 mV, as reviewer 2 suggested these experiments in his/her comment 3. Knocking out PKN1 and intracellular loading of TBOA in wild-type mice prolonged the decay time constant of NMDAR-EPSCs, suggesting that glutamate

uptake is reduced in PKN1a mice. The data on this finding are presented in Figure 6.

(7). Fig. 4 (= Fig. 7 in the present manuscript). I am not clear why PRL was tested only in WT, but not in KO, and the Dyn inhibitor tested only KO, but not in WT mice?

[Answer]

PRL, a PKN inhibitor, was loaded into a CA1 neuron, as shown in Fig. 7, to examine whether PKN regulates mGluR-LTD at postsynaptic sites. Therefore, we used hippocampal slices from wild-type mice. If we apply PRL to slices from PKN1a KO mice, we assumed that PRL does not affect mGluR-LTD. Dyn, a dynamin inhibitor, was used to examine whether mGluR-LTD in PKN1a KO mice involves endocytosis. Endocytosis of AMPA receptors has been well addressed in mGluR-LTD in the normal hippocampus (e.g., Xiao et al., *Neuropharm.*, 2001; Wilkerson et al., *Semin. Develop. Biol.*, 2018); therefore, we did not apply dyn to the wild-type hippocampus.

(8). Fig 5 and 6 (= Fig. 8 and 9 in the present manuscript). The failure rates under basal conditions in Fig 5 was the same between WT and KO mice, then why they were different in Fig. 6 between genotypes? Was the age of the mice in these two experiments different? More silent synapses in the KO mice suggest reduced synaptic AMPA receptors. Some biochemical/staining experiments would be helpful to confirm this.

[Answer]

(age)

We used wild-type and KO mice in the same age range (P8–12) in experiments (Figs. 8 and 9). Therefore, the differences in failure rate at -70 mV between Figs. 8 and 9 were not caused by the age of these mice (please see their figure legends).

(failure rate)

When we carried out minimal stimulation experiments, initially we roughly adjusted the failure rate at -70 mV to a desirable value for the purposes of the experiment by changing the stimulus intensity, although it was not always what we wanted. When we recorded LTD in experiments (Fig. 8), we hoped to have a slightly lower failure rate at -70 mV, because we expected an increase in the failure rate after LTD induction. On the other hand, when we calculated the percentage of silent synapses in Fig. 9, we adjusted the initial failure rates at -70 mV to a slightly higher value, especially in KO mice, because we were unable to calculate the percentage of silent synapses ($1 - \ln(F_{-70}) / \ln(F_{+40})$) if $F_{+40} = 0$. It could happen in KO mice; for example, a cell from a KO mouse shown in Fig. 9a had only two

failures at +40 mV. If we stimulated the afferents with a little higher intensity than we actually did and if the cell had no failures at +40 mV, we should have discarded the cell. In contrast to the frequency of miniature events (mEPSCs or mIPSCs), which have a fixed frequency in a cell, the failure rate in a cell can be changed depending on the stimulus intensity. Therefore, we can compare paired values of failure rates (i.e., failure rates of baseline and those at 30 min after LTD induction within wild-type or KO mice, in Fig. 8d); however, we cannot statistically compare failure rates between different cells (i.e., original failure rate at -70 mV between wild-type and KO mice, Fig. 8d, Fig. 9b). Thus, differences in the initial failure rates at -70mV between cells in Figure 8d and cells in Figure 9b do not reflect differences in cell properties.

(staining experiments)

We also think that staining AMPARs using culture cells from wild-type and KO mice is an interesting idea. In this study, we focused on data obtained from hippocampal slice experiments. Therefore, we hope to perform such staining experiments in our next project.

(9). Fig 7 (= Fig. 10 in the present manuscript). Model. Should the KO spine smaller??

[Answer]

In accordance with the comment of reviewer 3, we have reduced the size of the spine in KO mice (page 63). The size of spines in PKN1a KO mice (width, $0.514 \pm 0.008 \mu\text{m}$; length, $0.955 \pm 0.018 \mu\text{m}$) was 88% of that in wild-type mice (width, $0.583 \pm 0.008 \mu\text{m}$; length, $1.036 \pm 0.016 \mu\text{m}$), and the length of PSD in KO mice ($0.225 \pm 0.005 \mu\text{m}$) was 63% of that in wild-type mice ($0.358 \pm 0.008 \mu\text{m}$). Therefore, we illustrated the KO spine with a size of 88% of the wild-type spine and PSD in KO mice with a length of 63% of that in wild-type mice in Fig. 10.

REVIEWERS' COMMENTS:

Reviewer #1 (Remarks to the Author):

the authors did a very good job revising the manuscript and addressing reviewers' points. Most of my points have been addressed satisfactorily. I do not have any additional questions and recommend an acceptance of this manuscript.

Reviewer #2 (Remarks to the Author):

The authors have convincingly addressed my comments in the revised version of the manuscript.

Minor points:

- 1) line 183: "significantly different from APV only at $p = 0.0000000$ ": this is unclear.
- 2) line 239: I suggest to replace with "because inhibition of neuronal glutamate transporter was previously shown to prolong..."
- 3) line 342: "has less capacity..."
- 3) line 377: "have argued"
- 4) line 379 : "were"
- 5) Figure 3c and 5d "EPSP" instead of "EPEP"
- 6) Figure 5a c and d: "vehicle"

Corentin Le Magueresse

Reviewer #3 (Remarks to the Author):

I appreciate the authors' efforts to revise the paper extensively. Most of my comments have been addressed and therefore the paper is significantly improved. I have a couple of remaining comments that the authors may consider, but these do not need to be addressed experimentally.

- (1). It's interesting that mGluR antagonists plus APV did not block the PP-LFS-LTD in the KO mice completely. Do the authors know if the same concentrations block PP-LFS-LTD in WT?
- (2). I recognize authors' rationale for not doing the experiments for the PRL inhibitor in mutant mice and the dynamin inhibitor Dyn in WT, but these would be useful control experiments.

Dear reviewers,

We thank reviewers again for reviewing our revised manuscript and helpful comments. In accordance with the requests and suggestions from the editorial office of the journal, we have made changes to the manuscript as follows.

(Title)

We have changed the original title "PKN1 critically promotes synapse maturation by inhibiting mGluR-dependent silencing through neuronal glutamate transporter activation" to "PKN1 promotes synapse maturation by inhibiting mGluR-dependent silencing through neuronal glutamate transporter activation".

(Abstract)

We have shortened the original abstract (171 words), because the abstract is limited to 150 words long. We think the contents of the abstract was not significantly changed.

(original Figure 10)

Original Figure 10 (summary illustration of wild-type and KO spines) has been combined with Figure 9 and become Figure 9c. Accordingly, Fig. 10 has been changed to Fig. 9c in the manuscript (page 19, line 329; page 20, line 341).

Answers to comments from Reviewer #2:

Minor points:

1) line 183: "significantly different from APV only at $p = 0.0000000$ ": this is unclear.

[Answer]

We used EZR (easy R), a Japanese version of "R" (ref. 68), to perform multiple comparisons and it calculates the p values to 7 decimal places, and EZR does not show the exact value when p is smaller than 0.0000001. We also used SigmaPlot ver. 14 for multiple comparisons and it just calculated the p value to 4 decimal places. We are afraid that there is no available software that calculates p value to more than 7 decimal places, therefore, we have changed " $p = 0.0000000$ " to " $p < 0.0000001$ " (page 10, lines 179–180, and page 11, line 183).

2) line 239: I suggest to replace with "because inhibition of neuronal glutamate transporter was previously shown to prolong..."

[Answer]

In accordance with the suggestion of reviewer 2, we have changed “because inhibition of neuronal glutamate transporters by soluble amyloid β oligomers prolongs the rise time...” to “because inhibition of neuronal glutamate transporters by soluble amyloid β oligomers was previously shown to prolong the rise time” (page 14, lines 239–240).

3) line 342: "has less capacity..."

[Answer]

We have changed “...and have less capacity...” to “...and has less capacity...” (page 20, line 342)

4) line 377: "have argued"

[Answer]

We have changed “..have argue that....” to “...have argued that...” (page 22, line 371).

5) line 379 : "were"

[Answer]

We have changed “...fewer pulses of LFS was applied...” ...to “fewer pulses of LFS were applied...” (page 22, line 379).

6) Figure 3c and 5d "EPSP" instead of "EPEP"

7) Figure 5a c and d: "vehcle"

[Answer]

In accordance with the suggestions of reviewer 2, we have changed “EPEP (%)” to “EPSP (%)” in Y axis labels of graphs in Figs. 3c and 5d. Also, we have changed “vehcle” to “vehicle” in Figs. 5a, c, and d.

Answers to comments from Reviewer #3:

(1). It's interesting that mGluR antagonists plus APV did not block the PP-LFS-LTD in the KO mice completely. Do the authors know if the same concentrations block PP-LFS-LTD in WT?

[Answer]

We found that PP-LFS was a very intensive induction protocol for LTD and the amplitude of homosynaptic LTD in the presence of D-APV in wild-type mice was almost the same as that in the presence of D-APV and mGluR1 and 5 antagonists in KO mice (Fig. 4b). Therefore, we guess that application of D-APV and mGluR1 and 5 antagonists may not completely suppress homosynaptic LTD in wild-type mice, although we have not examined this.

(2). I recognize authors' rationale for not doing the experiments for the PRL inhibitor in mutant mice and the dynamin inhibitor Dyn in WT, but these would be useful control experiments.

[Answer]

We agree with the suggestion of reviewer 3. We had not carried out these control experiments just because our response to the reviewers' comments was substantially delayed and we wanted to hurry it. We thank the comment.